ecology

diet, ecology, elemental composition, *Gambusia affinis*, morphology, mosquitofish

**Author for correspondence:**
E. R. Moffett
e-mail: moffettemma@gmail.com

# Consumer trait responses track change in resource supply along replicated thermal gradients

E. R. Moffett[1], D. C. Fryxell[2], F. Lee[1], E. P. Palkovacs[2] and K. S. Simon[1]

[1]School of Environment, The University of Auckland, Private Bag 92019, Auckland, New Zealand
[2]Department of Ecology and Evolutionary Biology, University of California, Santa Cruz, CA 95060, USA

 ERM, 0000-0001-9891-6217; DCF, 0000-0003-4543-4809; FL, 0000-0002-9219-1486;
EPP, 0000-0002-5496-7263; KSS, 0000-0002-9537-2450

Rising temperatures may alter consumer diets through increased metabolic demand and altered resource availability. However, current theories assessing dietary shifts with warming do not account for a change in resource availability. It is unknown whether consumers will increase consumption rates or consume different resources to meet increased energy requirements and whether the dietary change will lead to associated variation in morphology and nutrient utilization. Here, we used populations of *Gambusia affinis* across parallel thermal gradients in New Zealand (NZ) and California (CA) to understand the influence of temperature on diets, morphology and stoichiometric phenotypes. Our results show that with increasing temperature in NZ, mosquitofish consumed more plant material, whereas in CA mosquitofish shifted towards increased consumption of invertebrate prey. In both regions, populations with plant-based diets had fuller guts, longer relative gut lengths, superior-orientated mouths and reduced body elemental %C and N/P. Together, our results show multiple pathways by which consumers may alter their feeding patterns with rising temperatures, and they suggest that warming-induced changes to resource availability may be the principal determinant of which pathway is taken.

## 1. Introduction

Higher temperature is predicted to elevate consumer metabolic demand with important consequences for consumer feeding and food web dynamics [1]. In warmer environments, increased metabolic demand may be met by increasing the overall consumption rate of the same foods or shifting diets towards foods that better satisfy the increased energy demand of metabolism [2–4]. For example, at warmer temperatures, the aquatic omnivorous pond snail (*Lymnaea stagnalis*) consumed greater amounts of plant material [3]. The driver of shifts in diet with temperature change could be metabolic and stoichiometric constraints that shift consumption to foods with higher carbon : phosphorus ratios at higher temperatures [5,6]. Alternatively, consumers may shift away from a plant-rich diet towards an animal-rich diet due to the energetic inefficiency of digesting plant materials [7–9].

Current theories on dietary shifts with rising temperature rest on shifting consumer demand from a physiological response to temperature, but they do not consider other changes in food webs that also occur with warming. Importantly, the temperature rise will change resource availability as thermal limits for some species are exceeded, and colonization by new species occurs [10–12]. As such, the dietary response of consumers is likely to depend on how temperature affects resource availability in addition to their physiological demand [7–10]. If animal-based resources are maintained or increase with warming (e.g. [13]), then consumers may increase their selectivity for animal-based prey. Alternatively, if animal-based resources decline with increased temperature (e.g. [14,15]), consumers may be forced to consume plant material and increase their overall consumption.

**Table 1.** Summary of the functional morphometric and stoichiometric traits measured on *Gambusia affinis* in this study and their functions (interpretations from [24–26,29,32]).

| trait | formula | function | implications for resources |
|---|---|---|---|
| relative gut length | $\dfrac{\text{gut length}}{\text{standard body length}}$ | energy-poor resources are associated with longer gut lengths | longer guts are associated with algal resources |
| mouth position | $\dfrac{\text{mouth opening}}{\text{head depth}}$ | feeding position in the water column | sub-terminal (lower) positioned mouths are associated with feeding on the benthos (e.g. Chironomidae), whereas superior (higher) mouths are associated with feeding on the surface or in the pelagic zone (e.g. filamentous algae, zooplankton) |
| eye size | $\dfrac{\text{eye diameter}}{\text{head depth}}$ | larger eyes for visual acuity | increased eye size enhances prey detection (e.g. animal resources) |
| eye position | $\dfrac{\text{eye height}}{\text{head depth}}$ | vertical position in the water column | lower eye position should favour feeding on the benthos |
| % body carbon | n/a | associated with lipid storage | increased temperature may lower body condition and therefore lipid stores |
| % body nitrogen | n/a | associated with muscle tissue | increased temperature may lower body condition and therefore muscle tissue |
| % body phosphorus | n/a | associated with skeletal structures | bony structures should remain the same with increased temperature, but N : P and C : P ratios may decline |

Shifts in resource availability and diet along thermal gradients should select for morphological traits which maximize resource use efficiency [16–20]. For example, plant-rich diets are associated with longer gut lengths due to higher food intake rates [21,22]. Organisms with plant-based diets may also compensate by increasing feeding rates (compensatory feeding) to meet their energetic demands [23]. Other traits related to feeding, such as eye position and size and mouth position, may also shift to optimize foraging [24]. For example, sub-terminal positioned mouths are useful to feed on the benthos, whereas superior orientated mouths are useful for surface feeding (e.g. upturned mouths of Poeciliidae) [25]. Thus, if metabolic demand and resource availability change along thermal gradients, the resulting changes in resource use may be associated with adaptive changes to morphological traits.

Body morphology and resource availability changes associated with increased temperatures may alter consumer elemental composition [26–28]. Where food availability is low, lipid storage (primarily C) is expected to decline while bony structure (P) requirements remain the same, lowering %C [29]. Alternatively, where animal resources are abundant, muscle tissue (primarily N) and lipid storage may increase, increasing %N and %C, respectively [30]. Variation in stoichiometry may also arise through altered stoichiometric requirements due to morphology. For example, if morphology requires increased bony structures, consumers may increase dietary *p* skeletal stores [26]. In addition to effects of resource availability and stoichiometric requirements, increased temperatures may influence elemental composition through increased energetic demand, lowering body condition via a reduction in lipids (%C) or muscle tissue (%N) if this demand is not met [6,29,31]. Therefore, if resource use and

morphology change along temperature gradients, this may also be reflected in consumer body stoichiometry.

Here, we use populations of western mosquitofish (*Gambusia affinis*) established from repeated recent invasions of springs that span parallel temperature gradients in New Zealand (NZ) and California (CA). We use a space for time substitution approach to explore how shifts in diet across the temperature gradients are associated with divergence in phenotypic traits (gut length, morphology) and body elemental composition. We hypothesized that with warmer temperatures, populations of *Gambusia affinis* would consume more plant-based materials as invertebrate resources decline and demand for carbon increases leading to longer guts, altered feeding traits (e.g. benthic feeding should align with a sub-terminal mouth position), and a body elemental composition reflective of environmental resources and morphology (table 1). Our aims were to (i) understand how resource availability and consumer diets shift with rising temperature, (ii) determine whether patterns of association with temperature are consistent across replicated introductions in NZ and CA, and (iii) describe the relationship between dietary change, gut morphology and body nutrient stoichiometry.

## 2. Methods

### (a) Study organism and populations

*Gambusia affinis* (hereafter '*Gambusia*') were introduced from populations in Texas, USA to CA, USA in the 1920s and to the North Island of NZ in the 1930s [33–35]. *Gambusia* are live-bearers, reach high densities in the wild, and are found across a wide range of environmental conditions: salinity, temperature, pH and dissolved oxygen (DO) [36]. *Gambusia* have a broad

**Table 2.** Characteristics of study sites in NZ and CA. DO and specific conductivity measurements were taken at the time of fish collection. Temperature is included at the time of fish collection and as an average value from seasonal point measures or temperature loggers. Pearson correlation values among site physiochemical variables are provided in electronic supplementary material, table S1.

| region | site | collection temperature (°C) | temperature average (°C) | DO % | specific conductivity (mS cm$^{-1}$) | pH | other large vertebrates |
|---|---|---|---|---|---|---|---|
| CA | WW5 | 18.8 | 20.4 | 126 | 0.492 | N/A | tui chub |
| | NE | 20.9 | 18.9 | 121 | 0.339 | 8.3 | none |
| | AW | 23.7 | 23.7 | 54 | 0.391 | 7.4 | none |
| | BLM | 24.0 | 21.1 | 98 | 0.156 | 8.2 | pupfish |
| | WSU | 27.8 | 27.4 | 82 | 0.461 | 7.8 | bullfrogs |
| | HC | 29.9 | N/A | 69 | 0.391 | N/A | unknown |
| | FC | 33.4 | N/A | 131 | 0.962 | N/A | bass |
| | LHC | 36.7 | 33.3 | 36 | 0.452 | 8.2 | none |
| | K2 | 38.9 | 31.6 | 118 | 0.827 | 8.4 | none |
| NZ | PP | 19.2 | 18.8 | 114 | 0.201 | 8.6 | common bully |
| | AL | 22.7 | N/A | 93 | 0.145 | 8.2 | none |
| | AD | 23.4 | 16.4 | 85 | 0.344 | 8.0 | goldfish |
| | PK | 24.0 | N/A | 31 | 3.566 | 7.4 | yellow-eyed mullet |
| | MR | 30.9 | 33.1 | 76 | 0.74 | 8.2 | none |
| | AA | 33.0 | 28.8 | 90 | 0.473 | 7.6 | guppies |
| | WA | 33.5 | 35.5 | 61 | 1.092 | 7.0 | none |
| | SP | 35.0 | 31.7 | 102 | 0.424 | 7.5 | none |
| | AWK | 37.7 | 36.4 | 87 | 0.391 | 9.0 | goldfish |

thermal niche, tolerating temperatures from freezing to approximately 40°C and reproduce in temperatures above approximately 16°C [36]. These omnivorous fish feed on algae, detritus, zooplankton, invertebrates and fish, and show a preference for consuming invertebrate foods when available [37–40].

We collected *Gambusia* from sites along parallel temperature gradients in NZ (19.2–37.7°C) and CA (18.8–38.9°C) (table 2). We studied 18 populations of *Gambusia* in soft-bottomed, slow-flowing geothermal spring systems in NZ and CA. Temperature, specific conductivity and DO were measured at each site using YSI ProODO and YSI Professional Plus metres. In both regions, temperature was not significantly correlated with DO (%), specific conductivity (mS cm$^{-1}$) and pH (electronic supplementary material, table S1). The springs were not chemically extreme, with conductivity values typical of non-geothermal waters in the regions, circumneutral pH and high levels of DO. The sites ranged from 1 to 263 km apart within each region and were not hydrologically connected on the surface (electronic supplementary material, figure S1 and table S2). Temperature differences among these sites were not associated with spatial distance among sites (electronic supplementary material, table S3). The presence of potential competitor and predator species was noted from visual surveys and netting carried out during site visits. Predator and competitor species were present in some sites (electronic supplementary material, table S4), but most sites lacked heterospecific fishes, and their presence was not correlated with temperature (table 2; electronic supplementary material, S1). Previous work on *Gambusia* in these geothermal systems demonstrated that metabolic and life-history traits have changed in response to temperature since the populations were established less than 100 years ago [41,42].

## (b) Sampling

In CA, we sampled between 30 May and 1 June 2016, and in NZ, we sampled from 8 February to 21 February 2017. At each site, *Gambusia* were captured using a 5 m seine (1.6 mm mesh) deployed at several locations. All *Gambusia* were immediately euthanized with MS-222 in CA or clove oil in NZ and transported back to the laboratory on ice and immediately frozen.

Macroinvertebrate communities were surveyed at each of the sites via a standard protocol using a D-net (0.5 mm mesh) [43]. We sampled ten 0.5 m$^2$ areas at each site, including different habitat types in proportion to their relative abundance. The contents of the D-net were pooled and preserved in 80% ethanol. Macroinvertebrates were identified to the lowest practical taxonomic unit (typically genus or family) under 10–80× magnification and counted to give relative abundances of each taxon and taxa richness in each sample. Invertebrates were pooled and dried at 60°C for 48 h to measure overall invertebrate dry mass. Planorbid snails were excluded from dry mass measurements as these are too large to be consumed by *Gambusia*.

Zooplankton were collected using a 63 µm mesh Wisconsin plankton net. At each site, this net was dragged horizontally under the water surface for 20 m, for a total volume of 13 l. All plankters were preserved in 80% ethanol. In the laboratory, zooplankters were enumerated and identified to the lowest possible taxonomic unit (typically genus or order) under a 10–80× magnification microscope.

## (c) Sample preparation

We randomly selected 40 (20 males and 20 females, where possible) mature individual *Gambusia* from each of the 18 sites ($n = 720$). *Gambusia* were weighed (±1 mg), and lateral photographs were taken for body morphological measurements. We

then removed the whole intestine of each *Gambusia*, cutting directly above the anus and immediately below the pharynx, uncoiled it and photographed it for length, and preserved it in 70% ethanol until gut content analysis. No other organs (e.g. liver, gall bladder) were removed during dissection. *Gambusia* (less their guts) were dried at 60°C for 48 h before being ground for whole-body stoichiometric analysis (see below).

We measured the functional morphological traits relative gut length, eye size, eye height and mouth position on each *Gambusia* using ImageJ [44]; these measurements are described in table 1.

## (d) Gut content analysis

Using the entire gut, individual gut contents were removed and placed onto a Petri dish with a 1 × 1 mm graticule. Contents were identified under an 80–100× microscope to the lowest taxonomic resolution possible. The proportion that each prey item contributed to total gut volume was visually estimated [45]. The volume of gut contents was quantified by gently pressing a microscope coverslip over the gut contents to a uniform depth of 1 mm$^2$ and counting how many 1 mm$^2$ cells on the graticule were filled [46]. Animal material (e.g. partially digested, unidentifiable invertebrate pieces) that could not be identified was summed into the 'amorphous animal' category. Plant material pre-digested by microbes in the environment or partially digested by fish was categorized as 'detritus'.

## (e) Nutrient stoichiometry

Body elemental composition was measured on 20 mature individuals (10 males and 10 females) from each of our 18 sites ($n = 360$). Elemental C and N were measured using a vario EL cube elemental analyser. For each individual, we used approximately 5 mg of dried and ground tissue (Elementar, Germany). For total P analysis, approximately 2–3 mg of dried and ground *Gambusia* tissue was ashed in a furnace at 550°C for 4 h. Combusted samples were digested by adding 10 ml of distilled water and 2 ml of 2 N HCl into each tube; tubes were then placed into an oven at 105°C for 2 h [47]. Following digestion, 0.5 ml of each sample was removed for spectrophotometric analysis according to the ascorbic acid method [48].

## (f) Statistical analysis

### (i) Model comparison

We used a model comparison approach to evaluate which factors (temperature, DO, specific conductivity, competitor species and spatial variation) best-explained variation in traits (e.g. gut length) among *Gambusia* populations in NZ and CA (electronic supplementary material, table S5). Only one site (FC in CA) had a known predator species (table 2). To simplify analyses, we removed this site before model comparison. Spatial variation was quantified using the *dbmem* function in 'adespatial' version 0.3–14 to compute distance-based Moran's eigenvector maps from a spatial distance matrix [49]. We constructed six models for each trait, one for each of the five factors mentioned above and a null model. For all models, we included a sex interaction, as *Gambusia* traits frequently varied with sex. We used the Akaike information criterion (AIC) to compare models using the *aictab* function in 'AICcmodavg' version 2.3-1 [50]. We ranked models by conditional Akaike information criterion (AICc) values and considered any models with ΔAICc less than four comparable in explanatory power (see electronic supplementary material, tables S6 and S7) [51].

### (ii) Diet and morphology

Variation in diet items (*i*) was summarized for each population using a relative importance index (RI$_i$). This index accounts for both frequencies of occurrence of a diet item (%F$_i$) and percentage volume in the gut (%V$_i$) [52].

$$\text{RI}_i = \frac{(\text{Al}_i\ 100)}{\sum_{i=1}^{n} \text{Al}_i}, \tag{2.1}$$

where Al$_i$ was calculated as %F$_i$ × %V$_i$.

We summarized variation in diet among populations using ordination by non-metric multidimensional scaling (NMDS). NMDS was carried out on the RI$_i$ data for each site using the Jaccard distance metric with temperature fit as an environmental gradient onto the ordination.

To determine if there were differences in the volume of prey in *Gambusia* guts that were independent of gut length, we calculated gut fullness (GF) as

$$\text{GF} = \frac{V}{\text{GL}}, \tag{2.2}$$

where $V$ is the volume of food in the *Gambusia* gut (mm$^2$) and GL is gut length (mm).

Independent generalized linear models (GLMs) were fitted to data from NZ and CA to analyse trends in GF, with site temperature and sex as independent factors.

We used simple linear models to determine if trends in diet (RI$_i$) and morphological traits (relative gut length, eye size, eye position and mouth position) were related to site temperature. We separated all morphological trait analyses by sex due to sexual dimorphism in *Gambusia* [37].

For plots of RI$_i$ and temperature, we pooled the major dietary categories (plant-based: algae + detritus, or animal-based: invertebrates + amorphous animal material) for comparison of the dominant trends.

### (iii) Stoichiometry

We used separate GLMs by both region and sex to understand the effects of temperature and *Gambusia* mass on %C, %N, %P, C/N, C/P and N/P. Data were split by sex as there is some expectation that stoichiometry may vary with sex [30]. We used temperature and individual wet mass as our independent variables.

To understand if *Gambusia* condition changed with temperature, we calculated individual condition factor (CF) [53] as

$$\text{CF} = W \times \frac{100}{\text{SL}^3}, \tag{2.3}$$

where $W$ is wet mass (g) and SL (cm) is the standard length. Five outlying (greater than 4 s.d. from the overall average value) CF values were excluded from analysis. We used GLMs with data were split by region to understand the effects of temperature and sex on CF.

Where appropriate, data were log$_{10}$ transformed before analysis to meet the models' normality assumptions. All statistical analyses were performed using R v. 4.1.0, and NMDS ordinations were produced using 'Vegan' version 2.5-6 [54,55]. All plots were created in R using ggplot2 v. 3.3-0 [56].

# 3. Results

## (a) Macroinvertebrate and zooplankton availability

Macroinvertebrate dry mass decreased 27-fold from our coolest to our warmest site in NZ ($r^2 = 0.651$, $p = 0.005$; figure 1*a*), but there was no relationship between temperature and macroinvertebrate dry mass in CA ($p = 0.321$; figure 1*d*). Across all sites, macroinvertebrate dry mass was about 2 X higher in CA (av = 0.048 g m$^2$, max = 0.109 g m$^2$) compared to NZ (av = 0.022 g m$^2$, max = 0.058 g m$^2$) and total species richness across all sites was higher in CA ($n = 13$) compared to NZ ($n = 9$). In both regions, all but one invertebrate was benthic (Dytiscidae in CA and *Mesovelia* in

Proc. R. Soc. B **288**: 20212144

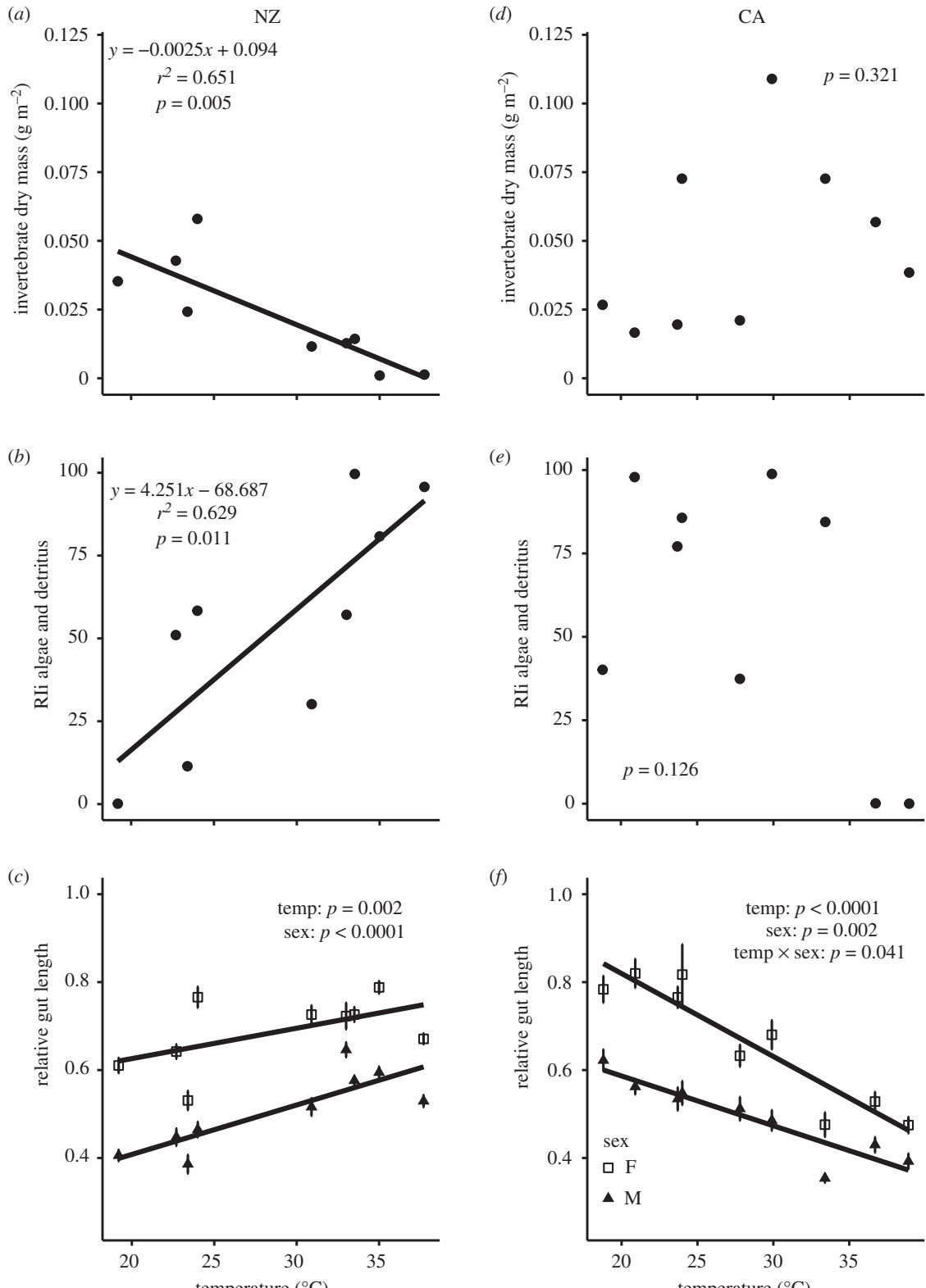

**Figure 1.** Relationship between temperature and site invertebrate dry mass, relative importance index scores (RIi) of algae and detritus in the diet of *Gambusia affinis*, and relative gut length of *Gambusia affinis*. Data are shown for NZ (*a,b,c*) and CA (*d,e,f*). Relative gut length data are averages ± s.e., split by sex. Invertebrate dry mass and RIi of algae and detritus data were fit with a linear regression model. Relative gut length data were modelled with a GLM (table 3). *n* = 9 per region.

NZ). Zooplankton density showed no relationship to temperature (*p* = 0.787) in NZ, but declined as site temperature increased (*r*² = 0.630, *p* = 0.017; electronic supplementary material, figure S2) in CA. Zooplankton density was 2.4 times lower across NZ sites when compared to CA. In both NZ (*p* = 0.524) and CA (*p* = 0.777), there were no distinct patterns of change in macroinvertebrate community composition with temperature (electronic supplementary material, figure S3).

## (b) Diet
In NZ and CA, there were shifts in diet at higher temperatures; however, these trends were opposite to one another and were stronger in NZ compared to CA (figure 1*b,e*; electronic supplementary material, figure S4). In NZ, RI*i* values shifted from invertebrate dominated to algal and detritus-dominated diet with temperature (*r*² = 0.629, *p* = 0.011), while we observed no significant trend in diet with

**Table 3.** GLM results describing the factors which explain relative gut length of *Gambusia affinis* in NZ and CA. Italics represent significance at $p < 0.05$.

| region | trait | coefficients | estimate | std. error | t | p |
|---|---|---|---|---|---|---|
| NZ | relative gut length | intercept | 0.425 | 0.072 | 5.884 | *<0.0001* |
| | | temp | 0.009 | 0.002 | 3.787 | *0.002* |
| | | sex | −0.180 | 0.030 | −6.065 | *<0.0001* |
| CA | relative gut length | intercept | 1.197 | 0.069 | 17.381 | *<0.0001* |
| | | temp | −0.019 | 0.002 | −7.947 | *<0.0001* |
| | | sex | −0.384 | 0.097 | −3.941 | *0.002* |
| | | temp × sex | 0.008 | 0.003 | 2.246 | *0.041* |

temperature CA ($p = 0.126$). Similarly, in NZ, NMDS ordination showed an increasing dominance of detritus, algae and terrestrial invertebrates in *Gambusia* diets with temperature ($r^2 = 0.762$; $p = 0.018$; electronic supplementary material, figure S2). In CA, NMDS trends were less clear, and differences were not related to temperature ($p = 0.576$).

## (c) Morphological traits and body chemistry

Modelling indicated that temperature was the best predictor of relative gut length and stoichiometric trends in CA and NZ (electronic supplementary material, table S5). Temperature was the best predictor of mouth position and one of the best predictors of eye size in CA. By contrast, in NZ, body morphological traits were most frequently explained by competitor presence. Specific conductivity best explained GF and CF in CA, whereas in NZ spatial distance best explained GF and DO, and specific conductivity best explained CF.

Females had longer relative gut lengths compared to males in NZ ($p < 0.0001$) and CA ($p = 0.002$), and within each region, both sexes showed similar patterns of gut length change with temperature. However, the direction of gut length change with temperature differed between regions (figure 1*c,f*). In NZ, where diet composition shifted to plant-based diets, relative gut length increased with temperature ($p = 0.002$; figure 1*c*; table 3). By contrast, in CA, where diet composition shifted towards animals, relative gut length decreased with temperature ($p < 0.0001$; figure 1*f*), with the degree of change being slightly stronger for females than males (temp × sex, $p = 0.041$).

While temperature and gut length trends were divergent, both regions showed similar relationships between gut length and plant material in *Gambusia* diet. In NZ, population average gut length increased with the relative importance of algae and detritus in *Gambusia* guts ($r^2 = 0.498$, $p = 0.034$; electronic supplementary material, figure S5*a*). There was a similar tendency for relative gut length to increase with algae and detritus in *Gambusia* diets in CA, although this trend was not statistically significant ($p = 0.235$; electronic supplementary material, figure S5*b*).

Body morphology measurements showed a divergence in mouth position with temperature between NZ and CA, whereas the presence of competitor species at our study sites did not significantly explain these measurements (electronic supplementary material, table S9). In CA, mouth position was lower at higher temperatures in males ($t = -5.73$, $p < 0.0001$) and females ($t = -5.54$, $p < 0.0001$; electronic supplementary material, table S8). Whereas in NZ, mouth position was higher with temperature, but this trend was significant for males ($t = 2.50$, $p = 0.013$) but not females ($t = 0.25$,

$p = 0.804$). In both regions, eye size increased with temperature. In CA, the increase in eye size with temperature was significant for males ($t = 3.72$, $p = 0.0003$) and females ($t = 3.79$, $p = 0.0002$). In NZ, this trend was significant for males ($t = 3.27$, $p = 0.001$) but not females ($t = 0.88$, $p = 0.381$). There were no significant trends in eye position in NZ or CA for either sex ($p > 0.05$, electronic supplementary material, table S8).

*Gambusia* exhibited a wide range of elemental variability with %C values ranging from 21.6 to 52.1, %N values ranging from 5.4 to 11.2 and %P values varying from 0.5 to 8.2 (electronic supplementary material, figure S6). Temperature was the key determinant of %C, %P, C/P and N/P in NZ, and %N and C/N in CA. In NZ and CA, temperature affected *Gambusia* elemental composition, where populations with plant-based diets had reduced body elemental %C and N/P. In NZ, %C (males: $t = -2.61$, $p = 0.011$; females: $t = -2.12$ $p = 0.037$) decreased with increasing site temperature (table 4). We found weaker trends for C/N (males: $t = -1.988$, $p = 0.050$; females: $t = -1.728$, $p = 0.088$) and %N (males: $t = 1.764$, $p = 0.081$), which decreased with increasing temperature in NZ. In NZ, male N/P decreased with increasing temperature ($t = -3.063$, $p = 0.003$), and there was a significant interaction between temperature and mass ($t = 2.194$, $p = 0.031$). For females, there was no relationship between N/P and temperature ($t = -1.521$, $p = 0.132$) or mass ($t = -0.725$, $p = 0.471$). There was no significant change in %P across the temperature gradient and dry mass was unrelated to elemental nutrient percentages ($p > 0.05$) in NZ. Whereas, in CA %C ($t = 1.979$, $p = 0.051$) and N/P increased ($t = 2.742$, $p = 0.007$) with increasing site temperature, but these patterns were only evident for males. In CA, %N increased with mass for females but not for males (females: $t = 2.067$, $p = 0.042$; males: $t = -0.837$, $p = 0.405$). For females in CA, there was an interaction between temperature and mass with %N ($t = -2.401$, $p = 0.019$), and there was a weak interaction between temperature and mass with N/P ($t = 1.971$, $p = 0.052$).

## 4. Discussion

Increased temperature is predicted to increase consumer energy demand and shift the composition of available resources, thereby altering consumer diets and potentially consumer trophic traits [10,12,20,57,58]. Our results suggest that increased temperature is associated with different changes in resource availability between regions. These resource changes, in turn, influence consumer diet, morphology and body elemental composition. However, thermal gradients in NZ and CA had

**Table 4.** GLM results describing which factors explain the nutrient percentages and ratios in fish body tissue of *Gambusia affinis* in NZ and CA. Mass is dry mass. Models with *p*—values < 0.10 are italics, and significance is noted as: '**' < 0.01, '*' < 0.05.

| region | sex | source | d.f. | %C | %N | %P | C/N | N/P | C/P |
|---|---|---|---|---|---|---|---|---|---|
| NZ | M | temp | 90 | −2.609* | −1.764 | 1.419 | −1.988 | −3.063** | 1.419 |
| | | mass | | −1.128 | −0.967 | 1.454 | −1.586 | −1.458 | 1.454 |
| | | temp × mass | | 1.279 | 1.927 | −1.286 | 1.498 | 2.194* | −1.286 |
| | F | temp | 90 | −2.115* | 0.153 | 1.185 | −1.728 | −1.521 | 1.185 |
| | | mass | | −0.158 | 1.040 | 0.550 | −0.594 | −0.725 | 0.550 |
| | | temp × mass | | −0.015 | −1.102 | −0.604 | 0.597 | 0.645 | −0.604 |
| CA | M | temp | 88 | 1.979 | −1.577 | −0.808 | 1.255 | 2.742** | −0.808 |
| | | mass | | 1.648 | −0.837 | −1.119 | 1.451 | 1.940 | −1.119 |
| | | temp × mass | | −1.214 | 0.455 | 0.854 | −1.095 | −1.312 | 0.854 |
| | F | temp | 88 | −0.139 | 0.465 | 0.716 | −0.780 | −0.267 | 0.716 |
| | | mass | | −0.306 | 2.067* | 1.272 | −1.351 | −1.396 | 1.272 |
| | | temp × mass | | 0.846 | −2.401* | −1.068 | 1.261 | 1.971 | −1.068 |

opposing trends in resource availability. Accordingly, resource use depends on community responses to increased temperature, leading to consistent changes in morphological divergence and body elemental composition among populations.

## (a) Temperature, resources and diets

Change in prey community composition and biomass is likely to occur with temperature change, as thermal limits for species are exceeded, and colonization by new species occurs [10–12]. Such change in prey resource availability across temperature gradients was crucial to interpreting *Gambusia* resource use differences in this study. In NZ, macroinvertebrate food resources were scarcer than in CA at cool temperatures and further decreased with increased temperature, probably necessitating a switch to a diet dominated by algal and detrital matter. Further, warm populations in NZ had a greater volume of food in their guts in comparison to cool populations, suggesting compensatory feeding with temperature rise to meet the higher metabolic energy demands associated with warmer temperatures [23]. Trends in CA were reversed, where invertebrate food resources remained abundant at the warmest sites, permitting diets dominated by macroinvertebrates and lower GF with higher temperature [9,59]. The stability of macroinvertebrate availability in CA may exist because of greater regional species pools in North America compared with the more isolated North Island of NZ. While it has been proposed that warming may lead to increased herbivory [3], and this is a common feeding strategy in many tropical fish species [60], our data and data from other studies indicate that this response may not be universal [9,61]. Overall, our data show multiple pathways by which *Gambusia* may alter their feeding patterns with rising temperatures, as *Gambusia* either forage for invertebrate prey or consume greater volumes of plant material to meet energy demands.

## (b) Morphological traits

Like other studies, our data show an increase in gut length with a shift toward plant-based diets [21,22,25,62–65]. While we found a significant relationship between relative gut length and algae and detritus in *Gambusia* diet in NZ, this relationship

was not significant in CA. Invertebrate resources were not as limited in CA, so *Gambusia* may be more reliant on invertebrate materials at all temperatures when compared to *Gambusia* in NZ, where animal resources were scarce at warm sites. The change in gut length with food resources suggests that *Gambusia* can adjust their phenotype to meet their energetic demands. Plasticity in gut morphology is common and may occur with diet changes or with fasting [22,62,65–67]. However, evolutionary change may occur concomitantly with plastic changes, particularly in spatially separated populations where there is potential for local adaptation. For example, Herrel *et al.* [68] found evolutionary divergence in lizard gut morphologies 36 years after introduction to novel environments. Similarly, diet manipulation experiments in Trinidadian guppies (*Poecilia reticulata*) did not alter the development of gut length, and gut length remained longest in populations with a plant-based diet, suggesting local adaptation [21]. The mechanism (i.e. evolution or plasticity) driving the trends in gut length in *Gambusia* here is unknown; however, the presence of parallel patterns of gut length with dietary shifts in NZ and CA highlights the functional significance of morphological changes in *Gambusia*.

The morphological traits, mouth position and eye size also responded to temperature. Mouth position responded in the same way to dietary shifts in NZ and CA, where mouth position became more superior (e.g. higher on the head) as diets became more plant-based, and mouth position became more sub-terminal (e.g. lower on the head) where diets were predominately animal-based. Many of the animal resources identified in our field sites were benthic; thus, our data suggest feeding on the benthos is a common feeding strategy for *Gambusia* where animal prey are abundant. When benthic animal prey were less available, mouth position was more superior. Long filamentous algae were common in warmer sites in NZ (E.R.M. 2017, pers. obs.), so fish may be feeding in the water column rather than on algae attached to the benthos. Alternatively, a superior mouth position may enhance surface feeding on terrestrial prey, which is typical of Poeciliidae [25].

In both regions, an increase in temperature was associated with larger eye size. In CA, increasing eye size with rising

temperature may enhance visual acuity for finding prey [24,69]. However, it is less clear why eye size increased with temperature in NZ, given that increasing temperature was associated with a reduction in invertebrate feeding. As such, other factors (e.g. flow, avoiding predators) may play a role in dictating morphological traits in NZ. Nevertheless, that eye size shifted consistently with temperature in both regions suggests that temperature may be a more important factor than resources for altering some morphological traits.

## (c) Elemental tissue composition

*Gambusia* elemental composition was influenced by both site temperature and sex. We found no other studies which have reported *Gambusia* elemental composition values; however, those found here were within the range of values found in other species in response to different biotic and abiotic factors [29,30,70,71]. Fish elemental composition shifted in response to temperature, but these trends were stronger for males compared to females. Differences may have been difficult to detect in females because of their greater trait variation. For example, female *Gambusia* have a longer lifespan than males, and *Gambusia* are sexually dimorphic (males grow very little after maturity, whereas females continue growing past maturity) [72–74]. Consequently, there is a greater range in body size in females which may alter lipid demand or bone formation through ontogeny. In addition, differences in stoichiometry may occur via difference in reproductive state and stage of embryo development [75]. Regardless of the mechanism, our data provide evidence that fish elemental composition varies with temperature and sex.

Dietary shifts towards plant and detrital material were consistently associated with increased N and reduced C in *Gambusia* elemental composition. While diet shifted in the opposite direction with temperature in NZ and CA, elemental composition tracked diet shift in the same manner. In NZ, %C, C/N and N/P decreased as temperature increased, and in CA, %C and N/P increased with temperature. Reduced %C and N/P with a plant-based diet in both regions indicates a reduction in lipid stores and muscle tissue, and thus an overall decrease in fish condition [21].

Despite the different patterns in stoichiometric proxies for condition between regions, our other condition metric—the CF—showed decreases with temperature in both regions. Although this pattern is typical of many fish species, including *Gambusia* [76], it should be noted that CF may serve as a poor proxy for overall condition. CF may not represent fat storage but may be more reflective of population differences in other traits like body shape. Together, these data suggest plant-based diets may lead to a reduction in overall fish condition via a reduction in body %C and N/P, and they suggest that changing resources may be a stronger determinant of stoichiometric changes along thermal gradients than temperature per se.

## 5. Conclusion

Using a space for time substitution approach, we observed that increased temperature was associated with altered resource availability. This was reflected in the diet of fish, with predictable changes in morphology and body elemental composition. Metabolic rates of mosquitofish are higher in the warmer systems [39], and there are alternative strategies for dealing with this increased energy demand: (i) eat increased quantities of plant material (eat more) or (ii) shift consumption towards increased proportions of invertebrate prey (eat better). Which of these strategies was employed depended on the response of the invertebrate prey community to increased temperature. As the temperature increased in NZ, invertebrate prey became less common, and *Gambusia* ate more algae. As the temperature increased in CA, invertebrate prey remained stable and *Gambusia* ate more animals. A plant-based diet was associated with comparatively longer and fuller guts, a more superior mouth position, and decreased elemental C and N/P in body tissue in both regions. Thus, to understand how consumer diet, elemental composition and morphological traits change with increased temperature, we need to understand how temperature influences resource availability and that may not play out the same in different places.

Ethics. Fish collection was approved by the University of California Santa Cruz (protocols PALKE-1311–2) and the University of Auckland animal ethics committees (Ref. 001089). In California, fish collection was also approved by the local wildlife agency (CADFW permit SC-12726). Fish collection in NZ was conducted with biosecurity permission from the Ministry for Primary Industries and with permissions from relevant Fish and Game Councils.

Data accessibility. Data are available from the Dryad Digital Repository: https://doi.org/10.7280/D1ZQ3C [77].

Authors' contributions. E.R.M.: conceptualization, data curation, formal analysis, funding acquisition, investigation, methodology, project administration, validation, visualization, writing-original draft, writing-review and editing; D.C.F.: conceptualization, investigation, methodology, validation, visualization, writing-review and editing; F.L.: conceptualization, data curation, investigation, methodology, writing-review and editing; E.P.P.: conceptualization, formal analysis, funding acquisition, investigation, methodology, resources, supervision, validation, visualization, writing-review and editing; K.S.S.: conceptualization, formal analysis, funding acquisition, investigation, methodology, resources, supervision, validation, visualization, writing-review and editing. All authors gave final approval for publication and agreed to be held accountable for the work performed therein.

Competing interests. We declare we have no competing interests.

Funding. This research was supported by the Royal Society of New Zealand Marsden Fund (16-UOA-23) and NSF (grant no. DEB 1457333). Partial support for E.R.M. was provided by the Kate Edger Educational Charitable Trust. Partial support for E.P.P. was provided by the NOAA Cooperative Institute for Marine Ecosystems and Climate.

Acknowledgements. For assistance with sample collection and processing, we thank Zachary Wood.

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
