## [Peer Review File · Proceedings of the Royal Society B: Biological Sciences]

Review History

RSPB-2020-1588.R0 (Original submission)

Review form: Reviewer 1

Recommendation

Major revision is needed (please make suggestions in comments)

Scientific importance: Is the manuscript an original and important contribution to its field?

Excellent

General interest: Is the paper of sufficient general interest?

Excellent

Quality of the paper: Is the overall quality of the paper suitable?

Good

Is the length of the paper justified?

Yes

Should the paper be seen by a specialist statistical reviewer?

No

Do you have any concerns about statistical analyses in this paper? If so, please specify them explicitly in your report.

No

It is a condition of publication that authors make their supporting data, code and materials available - either as supplementary material or hosted in an external repository. Please rate, if applicable, the supporting data on the following criteria.

Is it accessible?

No

Is it clear?

Yes

Is it adequate?

N/A

Do you have any ethical concerns with this paper?

No

Comments to the Author

This paper has the focus on two important aspects of our constantly changing world: invasive species and rising temperatures. As such, I think it will be of high general interest for a wide readership. Nonetheless, I have identified some aspects that the authors need to address as part of a revision.

Main comments:

1 - General: The authors begin by making a strong link towards increasing temperatures and how this will affect organisms. While I do not disagree with this link, I think it needs to be a bit more nuanced, and more carefully developed as the current data does not look at changing temperatures, but at different (potentially static) temperatures at discrete sites. These sites are likely to also differ in a whole slew of other biotic and abiotic variables besides temperature (please see additional points below), so the direct link from those sites to changing temperature on a wider scale is not very apparent to me in the current manuscript.

2 - Methods, lines 102-115: Much more information on the actual study systems is needed. How far are they apart, i.e., do they constitute true independent samples; also, a map - even if just for the supplementary material - would be extremely helpful. The authors cannot expect all readers to try and look up all the previous studies on these populations you cited. At the very least, I suggest adding latitude and longitude to Table 1. I also suggest to add a figure to the supplement that shows maps for the relevant regions in California and NZ, with markers identifying the location of each site. I tried getting the information I needed from some of the cited papers, but could not find all of the relevant information even in previous papers that are cited.

3 - Methods, lines 117-118: Based on this information here and in Table 1, these seem to be point measurements. How reliable are they relative to diel variation, or even weekly and monthly variation. Do the authors have longer-term data on these sites as well, that provide some evidence for temporal stability? Phenotypes of the fish the authors collected are likely to represent environmental conditions experienced over a longer period prior to collection. How can the authors be sure that their single-point measurements are properly representing these longer-term patterns?

4 - Methods, same lines as before: The authors mentioned at the beginning of the methods that these sites are all in geothermal springs. Do they have data on other water parameters in these aquatic habitats that could be due to different composition of discharge in each spring? I.e.,

amount of dissolved solids, minerals/salts, turbidity, any stressful compounds, etc.? Without this knowledge it will be difficult to really assess whether the uncovered patterns are really due to temperature or could be related to other, non-assessed differences between habitats. Following on from this, the authors further state that besides temperature, also conductivity and DO were quantified; yet, they seem not to have been included in any analytical models? I would like to see some rationale for this somewhere (I apologise if I missed it).

5 - Methods, lines 183-184: However, even relative gut length often differs between the sexes in *Gambusia*, other poeciliids and other fishes (e.g. Scharnweber et al. 2011 *Bull Fish Biol* 13:11-20, Näslund 2018 *Fishes* 3:38); in other words, even accounting for differences in body size between males and females, females may still have larger relative gut lengths. I therefore think sex will also have to be included as a factor in this analyses; or analyses should be split by sex.

6 - Discussion, lines 303-305: What about the role of competitors and predators? This also links back to my comment #1 on more detail on characteristics and location of each site. Are there any species of similar niche in these habitats, e.g. any pupfish in CA? Any native species in NZ that could influence *Gambusia* responses? Piscivorous fishes? If so, how could the presence or absence of competitors or predators have influenced dietary and morphological patterns (e.g., eye size discussion in lines 337-340). I think this needs to be more properly addressed.

7 - Discussion, lines 362-367: The condition factor only evaluates the weight/mass to length relationship, but not really fat storage. As such, you could have a situation in which fish have a different body shape (resulting in a different condition factor) but the same level of body fat. This needs to at least be acknowledged here.

Specific comments:

Introduction, line 61: Please delete the comma after 'If' at the beginning of the sentence.

Methods, line 149: I am not a native speaker but this does not sound correct. Did the authors mean 'The proportion that each'?

Ethics Statement: What about a sampling permit for NZ? I know that *Gambusia* are invasive species there, but one usually still needs an official permit to specifically collect them (even if just in case the collecting effort might influence native species adversely)?

Tables 3 and 4: Since the actual p-values are provided here, the stars denoting the level of significance are not needed and redundant.

References, lines 485-486: Information for reference #11 seems incomplete.

References, lines 510-512, 515-517, 563-564, 573-576: Please write out the journal name.

Review form: Reviewer 2

Recommendation

Reject – article is scientifically unsound

Scientific importance: Is the manuscript an original and important contribution to its field?

Acceptable

General interest: Is the paper of sufficient general interest?

Acceptable

Quality of the paper: Is the overall quality of the paper suitable?

Marginal

Is the length of the paper justified?

Yes

Should the paper be seen by a specialist statistical reviewer?

No

Do you have any concerns about statistical analyses in this paper? If so, please specify them explicitly in your report.

No

It is a condition of publication that authors make their supporting data, code and materials available - either as supplementary material or hosted in an external repository. Please rate, if applicable, the supporting data on the following criteria.

Is it accessible?

N/A

Is it clear?

N/A

Is it adequate?

N/A

Do you have any ethical concerns with this paper?

No

Comments to the Author

See attached (See Appendix A).

Decision letter (RSPB-2020-1588.R0)

21-Aug-2020

Dear Dr Moffett:

I am writing to inform you that your manuscript RSPB-2020-1588 entitled "Consumer trait responses track different resource changes along replicated thermal gradients" has, in its current form, been rejected for publication in Proceedings B.

This action has been taken on the advice of referees, who have recommended that substantial revisions are necessary. With this in mind we would be happy to consider a resubmission, provided the comments of the referees are fully addressed. In particular, please see the comments with regard to considering alternative variables other than temperature that may vary across your sites; we hope that you will provide additional data and analyses of these. Please note also that this is not a provisional acceptance.

The resubmission will be treated as a new manuscript. However, we will approach the same reviewers if they are available and it is deemed appropriate to do so by the Editor. Please note that resubmissions must be submitted within six months of the date of this email. In exceptional

circumstances, extensions may be possible if agreed with the Editorial Office. Manuscripts submitted after this date will be automatically rejected.

Yours sincerely,
 Professor Loeske Kruuk
 Editor
 mailto: proceedingsb@royalsociety.org

Associate Editor
 Board Member: 1
 Comments to Author:

This study investigates changes in diet, morphology, and elemental composition among populations of mosquitofish from two replicated temperature gradients. Both reviewers and myself found that the study's questions are interesting and the study to be well-conducted. Changes in diet in response to changes in climate, including temperature shifts, are critical to understand in our changing world, and data from natural systems are important.

Both reviewers raise important points that should be addressed.

First, the authors describe the observed variation in mosquitofish traits as a response to temperature. As reviewer 1 points out, other environmental variables are likely to vary among sites and to covary with temperature, so it is critical to evaluate some of these to allow a stronger assessment of the link to temperature. How do the conductivity and DO data that the authors collected covary with temperature and relate to the measured traits? Does the presence and abundance of predators and competitors vary among sites, and does this contribute to explaining trait variation? Does temperature have more explanatory power than a model of purely spatial variation among sites (temperature and space are likely to covary)? I would like to see the authors bring in additional data and modelling to address these possibilities.

Second, and related to the above point, more information is needed about the sites, as reviewer 1 notes. Is there any evidence that they are genetically differentiated populations? Without knowing this, it's hard to evaluate the correlations with temperature among sites. If closely-related and -located sites have been included, they will tend to cluster and might inflate the strength of relationships between fish traits and environmental variables. Is there additional temperature data beyond the authors' point measures to support the inference of long-term consistent temperature variation among sites?

Finally, the authors contrast the hypotheses that shifts in temperature will affect diet through affecting either organism metabolism or resource abundance. They conclude that changes in diet are more consistent with resource abundance shifts. But both hypotheses might be operating concurrently, so we need a discussion of whether mosquitofish metabolism might also shifted in response to temperature before concluding that it's mainly changing resource abundance that's important.

Reviewer 2 has suggested additional literature to set context and both reviewers have made additional suggestions that I hope the authors will find helpful.

Reviewer(s)' Comments to Author:

Referee: 1

Comments to the Author(s)

This paper has the focus on two important aspects of our constantly changing world: invasive species and rising temperatures. As such, I think it will be of high general interest for a wide readership. Nonetheless, I have identified some aspects that the authors need to address as part of a revision.

Main comments:

1 - General: The authors begin by making a strong link towards increasing temperatures and how this will affect organisms. While I do not disagree with this link, I think it needs to be a bit more nuanced, and more carefully developed as the current data does not look at changing temperatures, but at different (potentially static) temperatures at discrete sites. These sites are likely to also differ in a whole slew of other biotic and abiotic variables besides temperature (please see additional points below), so the direct link from those sites to changing temperature on a wider scale is not very apparent to me in the current manuscript.

2 - Methods, lines 102-115: Much more information on the actual study systems is needed. How far are they apart, i.e., do they constitute true independent samples; also, a map - even if just for the supplementary material - would be extremely helpful. The authors cannot expect all readers to try and look up all the previous studies on these populations you cited. At the very least, I suggest adding latitude and longitude to Table 1. I also suggest to add a figure to the supplement that shows maps for the relevant regions in California and NZ, with markers identifying the location of each site. I tried getting the information I needed from some of the cited papers, but could not find all of the relevant information even in previous papers that are cited.

3 - Methods, lines 117-118: Based on this information here and in Table 1, these seem to be point measurements. How reliable are they relative to diel variation, or even weekly and monthly variation. Do the authors have longer-term data on these sites as well, that provide some evidence for temporal stability? Phenotypes of the fish the authors collected are likely to represent environmental conditions experienced over a longer period prior to collection. How can the authors be sure that their single-point measurements are properly representing these longer-term patterns?

4 - Methods, same lines as before: The authors mentioned at the beginning of the methods that these sites are all in geothermal springs. Do they have data on other water parameters in these aquatic habitats that could be due to different composition of discharge in each spring? I.e., amount of dissolved solids, minerals/salts, turbidity, any stressful compounds, etc.? Without this knowledge it will be difficult to really assess whether the uncovered patterns are really due to temperature or could be related to other, non-assessed differences between habitats. Following on from this, the authors further state that besides temperature, also conductivity and DO were quantified; yet, they seem not to have been included in any analytical models? I would like to see some rationale for this somewhere (I apologise if I missed it).

5 - Methods, lines 183-184: However, even relative gut length often differs between the sexes in *Gambusia*, other poeciliids and other fishes (e.g. Scharnweber et al. 2011 Bull Fish Biol 13:11-20, Näslund 2018 Fishes 3:38); in other words, even accounting for differences in body size between males and females, females may still have larger relative gut lengths. I therefore think sex will also have to be included as a factor in this analyses; or analyses should be split by sex.

6 - Discussion, lines 303-305: What about the role of competitors and predators? This also links back to my comment #1 on more detail on characteristics and location of each site. Are there any species of similar niche in these habitats, e.g. any pupfish in CA? Any native species in NZ that could influence *Gambusia* responses? Piscivorous fishes? If so, how could the presence or absence of competitors or predators have influenced dietary and morphological patterns (e.g., eye size discussion in lines 337-340). I think this needs to be more properly addressed.

7 - Discussion, lines 362-367: The condition factor only evaluates the weight/mass to length relationship, but not really fat storage. As such, you could have a situation in which fish have a different body shape (resulting in a different condition factor) but the same level of body fat. This needs to at least be acknowledged here.

Specific comments:

Introduction, line 61: Please delete the comma after 'If' at the beginning of the sentence.

Methods, line 149: I am not a native speaker but this does not sound correct. Did the authors mean 'The proportion that each'?

Ethics Statement: What about a sampling permit for NZ? I know that *Gambusia* are invasive species there, but one usually still needs an official permit to specifically collect them (even if just in case the collecting effort might influence native species adversely)?

Tables 3 and 4: Since the actual p-values are provided here, the stars denoting the level of significance are not needed and redundant.

References, lines 485-486: Information for reference #11 seems incomplete.

References, lines 510-512, 515-517, 563-564, 573-576: Please write out the journal name.

Referee: 2

Comments to the Author(s)

See attached.

Author's Response to Decision Letter for (RSPB-2020-1588.R0)

See Appendix B.

RSPB-2021-0595.R0

Review form: Reviewer 1

Recommendation

Accept with minor revision (please list in comments)

Scientific importance: Is the manuscript an original and important contribution to its field?

Excellent

General interest: Is the paper of sufficient general interest?

Excellent

Quality of the paper: Is the overall quality of the paper suitable?

Excellent

Is the length of the paper justified?

Yes

Should the paper be seen by a specialist statistical reviewer?

No

Do you have any concerns about statistical analyses in this paper? If so, please specify them explicitly in your report.

No

It is a condition of publication that authors make their supporting data, code and materials available - either as supplementary material or hosted in an external repository. Please rate, if applicable, the supporting data on the following criteria.

Is it accessible?

Yes

Is it clear?

Yes

Is it adequate?

Yes

Do you have any ethical concerns with this paper?

No

Comments to the Author

I would like to thank the authors for their careful revision and for the way they addressed my comments from the previous round of review (e.g., by adding maps of the study region to the supplement). I am satisfied with these changes, but when working through the new version, I stumbled over a few additional points that will need to be addressed.

Specific points:

Abstract, line 20: Maybe this is because I am not a native speaker, but 'Temperature rise' sounds odd to me and I would rephrase this to 'Rising temperatures'. Also, I think it should be 'demands' rather than 'demand'.

Abstract, line 30: I found the wording 'longer gut length to body length ratios' a bit odd, and would simply refer to this as 'relative gut length' as is done throughout the main text.

Introduction, lines 45-50: Are the last two sentences supposed to explain the pattern found for *L. stagnalis* or refer back to the overall pattern? Currently, this is not fully clear from the wording. If the former, then the explanations seem to contradict the pattern reported as an example; if the latter, then please rephrase to make this connection more clear.

Methods, lines 111-120: Given that these patterns are based on the measurements taken as part of this study, this should all be in past tense.

Methods, line 127: Please change 'using' to 'via'

Methods, lines 141-142: Here and for the other sample sizes below: Were these a mix of adult and juvenile fish or were these all adults? This would be important to know as a mixing of juveniles and adults introduces additional noise to the dataset and could result in spurious significance when comparing samples that could solely be based on the proportion of juveniles in different samples. I am assuming from the SL values provided in the data that all were adults but this would be important to state specifically.

Methods, line 188: Please provide a relevant citation for the sexual size dimorphism.

Results, line 229: Please add wording that makes clear that you do not consider this significant - the associated p-value shows this but your wording seems to suggest you considered this on the same level as the significant opposite trend in NZ.

Results, line 259: Please add 'and' prior to '%P'.

Table 4: The addition of the full stop after the t-value is confusing, given that a full stop is already part of every t-value. I suggest to simply not use any symbol for the '<0.05' significance. The bold already indicates it was significant, so no additional symbol is needed to separate it from *, ** and ***.

Figure 1: Detritus is mis-spelled on the y-axes of panels b and e.

Review form: Reviewer 2

Recommendation

Accept with minor revision (please list in comments)

Scientific importance: Is the manuscript an original and important contribution to its field?

Good

General interest: Is the paper of sufficient general interest?

Good

Quality of the paper: Is the overall quality of the paper suitable?

Excellent

Is the length of the paper justified?

Yes

Should the paper be seen by a specialist statistical reviewer?

No

Do you have any concerns about statistical analyses in this paper? If so, please specify them explicitly in your report.

No

It is a condition of publication that authors make their supporting data, code and materials available - either as supplementary material or hosted in an external repository. Please rate, if applicable, the supporting data on the following criteria.

Is it accessible?

Yes

Is it clear?

Yes

Is it adequate?

Yes

Do you have any ethical concerns with this paper?

No

Comments to the Author

See attached.

Decision letter (RSPB-2021-0595.R0)

27-Apr-2021

I am writing to inform you that we now have reviews of your manuscript RSPB-2021-0595 entitled "Consumer trait responses track change in resource supply along replicated thermal gradients", and the paper has been evaluated by the Associate Editor.

Whilst the reviewers and the Associate Editor are all agreed that the revised version of the paper is considerably improved, there are still some substantial issues to be addressed. In particular, the AE raises important points about testing for the role of other variables in addition to temperature as the explanation for the observed changes. These will require reanalyses and additional data. The paper has therefore been rejected in its current form, but with the invitation to resubmit provided the comments of the Associate Editor and the referees are fully addressed. I must emphasise that we are all agreed that this is an interesting topic and valuable study systems, but you still need to convincingly address the concerns raised with the previous versions of the manuscript; this is not a provisional acceptance.

Please find below the comments made by the referees, not including confidential reports to the Editor, which I hope you will find useful.

- 1) A 'response to referees' document including details of how you have responded to the comments, and the adjustments you have made.
- 2) A clean copy of the manuscript and one with 'tracked changes' indicating your 'response to referees' comments document.

- 3) Line numbers in your main document.
 4) Please read our data sharing policies to ensure that you meet our requirements <https://royalsociety.org/journals/authors/author-guidelines/#data>.

We look forward to receiving your revised manuscript.

Yours sincerely,
 Professor Loeske Kruuk
 Editor
 mailto: proceedingsb@royalsociety.org

Associate Editor Board Member: 1

Comments to Author:

Both original reviewers and myself found the manuscript improved, and the additional information about site location, average temperature and other properties is helpful. However, it remains my view that additional data and modelling are needed before the results can be seen as a fully convincing case that temperature drives population variation in the mosquitofish traits studied here. First, a model comparison approach (e.g. based on Akaike information criterion) would help to evaluate whether a model including temperature has more explanatory power than a purely spatial model, or models based on pH, other chemical properties, or predators and competitors. A null model (e.g. that includes only the intercept) should also be included in the comparison set. The authors have introduced a Mantel test to show that temperature and spatial differences among sites are not strongly correlated, but this does not exclude the possibility that spatial variation explains trait variation better than temperature. Second, I suggest the authors consider an alternative spatial analysis to the Mantel test (see Legendre et al. 2015, Should the Mantel test be used in spatial analysis? *Methods in Ecology and Evolution* 6: 1239-1247). Third, I would like to see a more robust analysis of predator and competitor effects, as Reviewer 1 raised in their initial review. The revised manuscript introduces some data on predator and competitor presence or absence, but predators and competitors are grouped together without clear distinction and only one predator or competitor is listed for a subset of sites (surely some sites must have had more than one?). No source is given for these data and there are no data on predator or competitor abundance. The authors state that predator or competitor presence is not associated with temperature, but there probably isn't a good way to evaluate this given the current quality of the predator/competitor data.

Reviewer(s)' Comments to Author:

Referee: 1

Comments to the Author(s).

I would like to thank the authors for their careful revision and for the way they addressed my comments from the previous round of review (e.g., by adding maps of the study region to the supplement). I am satisfied with these changes, but when working through the new version, I stumbled over a few additional points that will need to be addressed.

Specific points:

Abstract, line 20: Maybe this is because I am not a native speaker, but 'Temperature rise' sounds odd to me and I would rephrase this to 'Rising temperatures'. Also, I think it should be 'demands' rather than 'demand'.

Abstract, line 30: I found the wording 'longer gut length to body length ratios' a bit odd, and would simply refer to this as 'relative gut length' as is done throughout the main text.

Introduction, lines 45-50: Are the last two sentences supposed to explain the pattern found for *L. stagnalis* or refer back to the overall pattern? Currently, this is not fully clear from the wording. If the former, then the explanations seem to contradict the pattern reported as an example; if the latter, then please rephrase to make this connection more clear.

Methods, lines 111-120: Given that these patterns are based on the measurements taken as part of this study, this should all be in past tense.

Methods, line 127: Please change 'using' to 'via'

Methods, lines 141-142: Here and for the other sample sizes below: Were these a mix of adult and juvenile fish or were these all adults? This would be important to know as a mixing of juveniles and adults introduces additional noise to the dataset and could result in spurious significance when comparing samples that could solely be based on the proportion of juveniles in different samples. I am assuming from the SL values provided in the data that all were adults but this would be important to state specifically.

Methods, line 188: Please provide a relevant citation for the sexual size dimorphism.

Results, line 229: Please add wording that makes clear that you do not consider this significant - the associated p-value shows this but your wording seems to suggest you considered this on the same level as the significant opposite trend in NZ.

Results, line 259: Please add 'and' prior to '%P'.

Table 4: The addition of the full stop after the t-value is confusing, given that a full stop is already part of every t-value. I suggest to simply not use any symbol for the '<0.05' significance. The bold already indicates it was significant, so no additional symbol is needed to separate it from *, ** and ***.

Figure 1: Detritus is mis-spelled on the y-axes of panels b and e.

Referee: 2

Comments to the Author(s).

See attached.

Author's Response to Decision Letter for (RSPB-2021-0595.R0)

See Appendix C.

RSPB-2021-0595.R1

Review form: Reviewer 1

Recommendation

Accept with minor revision (please list in comments)

Scientific importance: Is the manuscript an original and important contribution to its field?

Excellent

General interest: Is the paper of sufficient general interest?

Excellent

Quality of the paper: Is the overall quality of the paper suitable?

Excellent

Is the length of the paper justified?

Yes

Should the paper be seen by a specialist statistical reviewer?

No

Do you have any concerns about statistical analyses in this paper? If so, please specify them explicitly in your report.

No

It is a condition of publication that authors make their supporting data, code and materials available - either as supplementary material or hosted in an external repository. Please rate, if applicable, the supporting data on the following criteria.

Is it accessible?

Yes

Is it clear?

Yes

Is it adequate?

Yes

Do you have any ethical concerns with this paper?

No

Comments to the Author

Again, I would like to thank the authors for their careful revisions. I am now mostly satisfied and am pointing out the last few items that, in my opinion, should be corrected prior to final acceptance

Specific comments:

Lines 104-106: Another very recent diet study that could be cited here is Pirroni et al. 2021 *Ecol Evol* 11:4379–4398.

Supplementary Tables: Thank you for changing how significance is indicated in Table 4 (now Table 5). Please also change the designation of significance in all the relevant supplementary tables, as again, using a full stop after the p-value to indicate significance level is very confusing.

Review form: Reviewer 2

Recommendation

Accept as is

Scientific importance: Is the manuscript an original and important contribution to its field?

Good

General interest: Is the paper of sufficient general interest?

Good

Quality of the paper: Is the overall quality of the paper suitable?

Good

Is the length of the paper justified?

Yes

Should the paper be seen by a specialist statistical reviewer?

No

Do you have any concerns about statistical analyses in this paper? If so, please specify them explicitly in your report.

No

It is a condition of publication that authors make their supporting data, code and materials available - either as supplementary material or hosted in an external repository. Please rate, if applicable, the supporting data on the following criteria.

Is it accessible?

Yes

Is it clear?

Yes

Is it adequate?

Yes

Do you have any ethical concerns with this paper?

No

Comments to the Author

The authors responded adequately to all queries.

Decision letter (RSPB-2021-2144.R0)

02-Nov-2021

Dear Dr Moffett,

I am pleased to inform you that your manuscript RSPB-2021-2144 entitled "Consumer trait responses track change in resource supply along replicated thermal gradients" has been accepted for publication in Proceedings B.

The referees and Associate Editor have recommended publication, but have also suggested some minor revisions to your manuscript. Therefore, I invite you to respond to the referee's comments and revise your manuscript. Because the schedule for publication is very tight, it is a condition of publication that you submit the revised version of your manuscript within 7 days. If you do not think you will be able to meet this date please let us know.

- DNA sequences: Genbank accessions F234391-F234402

- Phylogenetic data: TreeBASE accession number S9123
- Final DNA sequence assembly uploaded as online supplemental material
- Climate data and MaxEnt input files: Dryad doi:10.5521/dryad.12311

[http://datadryad.org/submit?journalID=RSPB&manu=\(Document not available\)](http://datadryad.org/submit?journalID=RSPB&manu=(Document%20not%20available)) which will take you to your unique entry in the Dryad repository. If you have already submitted your data to dryad you can make any necessary revisions to your dataset by following the above link. Please see <https://royalsociety.org/journals/ethics-policies/data-sharing-mining/> for more details.

Yours sincerely,

Loeske Kruuk

Editor

Associate Editor

Board Member

Comments to Author:

The authors have handled the revisions carefully. The reviewers are satisfied with the authors' responses, as am I. The model comparison has been handled well and strengthens the interpretation. This revision is a well-presented manuscript on a very interesting question.

Reviewer(s)' Comments to Author:

Referee: 1

Comments to the Author(s).

Again, I would like to thank the authors for their careful revisions. I am now mostly satisfied and am pointing out the last few items that, in my opinion, should be corrected prior to final acceptance

Specific comments:

Lines 104-106: Another very recent diet study that could be cited here is Pirroni et al. 2021 *Ecol Evol* 11:4379–4398.

Supplementary Tables: Thank you for changing how significance is indicated in Table 4 (now Table 5). Please also change the designation of significance in all the relevant supplementary tables, as again, using a full stop after the p-value to indicate significance level is very confusing.

Referee: 2

Comments to the Author(s).

The authors responded adequately to all queries.

Author's Response to Decision Letter for (RSPB-2021-2144.R0)

See Appendix D.

Decision letter (RSPB-2021-2144.R1)

04-Nov-2021

Dear Dr Moffett

I am pleased to inform you that your manuscript entitled "Consumer trait responses track change in resource supply along replicated thermal gradients" has been accepted for publication in Proceedings B.

Data Accessibility section

Open Access

Paper charges

You are allowed to post any version of your manuscript on a personal website, repository or preprint server. However, the work remains under media embargo and you should not discuss it

with the press until the date of publication. Please visit <https://royalsociety.org/journals/ethics-policies/media-embargo> for more information.

Sincerely,
Editor, Proceedings B
mailto: proceedingsb@royalsociety.org

Appendix A

Review of “Consumer trait responses track different resource changes along replicated thermal gradients” by Moffett et al., a manuscript submitted to PRSB.

In this paper, the authors studied how *Gambusia affinis* responded to changing temperature and resource availability across thermal gradients in California and New Zealand. Thus, they represented replicate thermal gradient experiments with separate invasions of *G. affinis* in the two locations (*Gambusia* are not native to either location and had been in each for at least 85 years or so). They made some predictions about how temperature would affect diet (increased algal consumption with increased temperature) and how these changes would in turn affect body morphology (including mouth and eye shape, and gut length). Their hypotheses basically underscored that the fish would need more C due to higher metabolic rates at higher temperatures. Overall, I have no problems with the science itself. It is well-designed and executed. I do have problems with the scholarship, which is insufficient in many areas of the introduction. And, it is unfortunate that they could not replicate their findings in NZ in CA, showing that the pattern in NZ is not a universal one (mainly because the resource changes with temperature were not uniform among the locations). The variation in the responses by the fish are interesting in and of themselves, but I wonder if they are enough for PRSB. The introduction needs some work to match the citations with what is stated in the sentences. They don't always match up. The idea of complementarity isn't necessarily a new one, as omnivores are known to seek out C sources for energy, and more proteinaceous sources for N:

Raubenheimer, D., Zemke-White, W. L., Phillips, R. J., & Clements, K. D. (2005). Algal macronutrients and food selectivity by the omnivorous marine fish *Girella tricuspidata*. *Ecology*, 86, 2601-2610.

And, the idea that animals would eat more of a low-protein diet to meet their needs isn't entirely novel, either. So, I appreciate many aspects of this study and it was an amazing effort in multiple locations. I applaud all of that. But, the writing left me unsatisfied.

Below, I detail comments on many parts of the introduction where the citations don't match the information in the sentences.

Page 3, line 47: Here, the authors can cite McWilliams and Karasov (2014) Proc Royal Soc B, where they used cold temperature to increase metabolic demand in an endotherm (bird). The birds showed increased food intake and changes in gut structure and function (which is key since most of the other cited studies didn't look at changes to the digestive system).

McWilliams, S. R., & Karasov, W. H. (2014). Spare capacity and phenotypic flexibility in the digestive system of a migratory bird: defining the limits of animal design. *Proceedings of the Royal Society B: Biological Sciences*, 281(1783), 20140308.
doi:doi:10.1098/rspb.2014.0308

Page 3, line 50: The authors cite Floeter et al. as evidence of some temperature-based constraints on digestive physiology of herbivores. Note that Floeter et al. provide no evidence of any such constraints, and in fact, Dr. Kendall Clements (University of Auckland) has a research program

that includes many cold-water herbivores (e.g., *Odax pullus*) that have rates of gastrointestinal fermentation that rival endotherms at cooler temperatures (Mountfort et al. 2002). Not to mention, no impedance of growth (Trip et al. 2014) or diet shifts (Johnson et al. 2020) at higher latitudes. Antarctic omnivores graze on algae from the bottom of the ice (Iken et al. 1997), and there are cool temperate herbivores in the northern hemisphere that digest algae with high efficiency as well (Horn et al. 1986; German et al. 2015). So, the authors should re-think that part of this sentence. The larger patterns of lower diversity and abundance of herbivorous fishes at higher latitudes can have many explanations beyond a digestive constraint (e.g., reproductive limitations, or stoichiometric ones).

German, D. P., Sung, A., Jhaveri, P. K., & Agnihotri, A. (2015). More than one way to be an herbivore: convergent evolution of herbivory using different digestive strategies in pricklyback fishes (family Stichaeidae). *Zoology*, *118*, 161-170.

Horn, M. H., Neighbors, M. A., & Murray, S. N. (1986). Herbivore responses to a seasonally fluctuating food supply: growth potential of two temperate intertidal fishes based on the protein and energy assimilated from their macroalgal diets. *Journal of Experimental Marine Biology and Ecology*, *103*, 217-234.

Iken, K., Barrera-Oro, E. R., Quartino, M. L., Casaux, R. J., & Brey, T. (1997). Grazing by the Antarctic fish *Notothenia coriiceps*: evidence for selective feeding on macroalgae. *Antarctic Science*, *9*, 386-391.

Johnson, J. S., Raubenheimer, D., Bury, S. J., & Clements, K. D. (2020). Does temperature constrain diet choice in a marine herbivorous fish? *Marine Biology*, *167*(7), 99. doi:10.1007/s00227-020-3677-z

Trip, E. D. L., Clements, K. D., Raubenheimer, D., & Choat, J. H. (2014). Temperature-related variation in growth rate, size, maturation and life span in a marine herbivorous fish over a latitudinal gradient. *Journal of Animal Ecology*, *83*(4), 866-875. doi:10.1111/1365-2656.12183

Page 3, line 52: The authors state here that it could be C:N ratios that drive dietary changes with increasing metabolic rate, then cite two papers that looked at C:P (and number 9 specifically in animals with high P content), not C:N. Please find citations that correctly show this for C:N ratios.

Page 3, line 52: The authors state here that “Alternatively, consumers may shift away from a plant-rich diet towards an animal-rich diet due to the energetic inefficiency of digesting plant materials”. They cite a paper on carnivorous trout (4), a model (number 10), a review article (11) that doesn't really cover inefficient digestion of plant material in herbivores (in fact, it shows herbivores use plants pretty well), and an article that used synthetic diets with varying protein to carbohydrate ratios in grasshoppers. None of these citations match this sentence to show energetic inefficiency of digesting plant materials by animals that normally consume plants (although Pandas don't do much with the fiber portion of bamboo; Dierenfeld et al. 1982; Nie et

al. 2019). I ask the authors to dig and find such papers, because there are few (Floeter cites one on reptiles that is highly flawed).

Dierenfeld, E. S., Hintz, H. F., Robertson, J. B., Van Soest, P. J., & Oftedal, O. T. (1982). Utilization of bamboo by the giant panda. *Journal of Nutrition*, 112, 636-641.

Nie, Y. G., Wei, F. W., Zhou, W. L., Hu, Y. B., Senior, A. M., Wu, Q., . . . Raubenheimer, D. (2019). Giant Pandas Are Macronutritional Carnivores. *Current Biology*, 29(10), 1677-1682.e1672. doi:<https://doi.org/10.1016/j.cub.2019.03.067>

Page 4, line 67: Here, the authors state that “For example, plant-rich diets are associated with longer gut lengths within and across species due to the greater processing times required to process refractory materials in plants [11, 23, 24].” They cite a review and two papers that examine gut length in fishes, yet none of them measure retention time of food in the gut (e.g., see Horn 1989; German 2011 for discussions on this) or digestibility of refractory plant material. The authors imply that herbivores hold food in their guts longer and that increased retention time (i.e., decreased transit rate) is associated with longer guts to allow microbes to help in fiber digestion. This is not true across the board (Horn 1989; Clements et al. 2014), and there are few papers that actually measure fiber digestion in fishes (German 2009 is an example). A longer gut simply means higher intake (particularly in non-ruminant animals, which is most animals, and this is covered in Karasov and Douglas 2013 and Karasov and Martinez del Rio 2007. The authors already cite Leigh et al. 2018, which actually has a detailed discussion of this). Higher intake is associated with more rapid gut transit, not slower transit (German 2011), especially in thin walled intestines, like those of poeciliids. So, the authors here confuse “processing time” with intake. Unless the gut has some mechanism to slow transit (valves, ceca, compartmentalization), then longer guts usually mean rapid transit (and actually, less microbial involvement in the digestion of plant refractory material). Few fishes really have long transit and a reliance on microbes to digest “refractory” material in plants and algae. In fact, those fishes with rapid fermentation occurring in their hindguts may be targeting mannitol and not the more fibrous parts of algae (White et al. 2010; Clements et al. 2014). So, the authors cannot make this statement and cite papers that do not show what they say. Longer guts mean higher intake. That’s it. Herbivores can take many different strategies to digest plant material (German et al. 2015). Higher intake and longer gut usually means they are going after more soluble portions of the plant material and are passing more of it through their gut per unit time to meet their nutritional requirements (perhaps for protein; Horn et al. 1986; Clements and Raubenheimer 2006). Finally, no poeciliids have been shown to have much in the way of gastrointestinal fermentation occurring, so a longer gut in them likely just equates to higher intake and more rapid gut transit, as described above. The reported relative gut lengths are short for animals “reliant on plant material”. I share all of this detail because these are issues with the writing, not the science performed. So, I want the authors to understand why gut length is a different issue than what they shared. High intake is probably about meeting their protein requirement from a diet that is relatively low in it. Finally, I share all of these citations not for all of them to be cited, but for the authors to have access to the information to give context to the issue at hand.

Clements, K. D., & Raubenheimer, D. (2006). Feeding and nutrition. In D. H. Evans (Ed.), *The physiology of fishes* (pp. 47-82). Boca Raton, FL: CRC Press.

- Clements, K. D., Angert, E. R., Montgomery, W. L., & Choat, J. H. (2014). Intestinal microbiota in fishes: what's known and what's not. *Molecular Ecology*, 23(8), 1891-1898. doi:10.1111/mec.12699
- German, D. P. (2009). Inside the guts of wood-eating catfishes: can they digest wood? *Journal of Comparative Physiology B*, 179, 1011-1023.
- German, D. P. (2011). Digestive Efficiency. In A. P. Farrel (Ed.), *Encyclopedia of Fish Physiology: From Genome to Environment* (Vol. 3, pp. 1596-1607). San Diego: Academic Press.
- Horn, M. H. (1989). Biology of Marine Herbivorous Fishes. *Oceanography and Marine Biology Annual Review*, 27, 167-272.
- White, W. L., Coveny, A. H., Robertson, J., & Clements, K. D. (2010). Utilisation of mannitol by temperate marine herbivorous fishes. *Journal of Experimental Marine Biology and Ecology*, 391(1-2), 50-56. doi:http://dx.doi.org/10.1016/j.jembe.2010.06.007

Page 4, lines 73-75: are the authors suggesting that in a single species, these morphological changes (downward turned vs upward turned mouths) occur across a thermal gradient in the same species? Or that these changes can occur rapidly with climate change? It isn't clear.

Page 5, line 95: The authors need to cite Table 2 here so that the predictions are clear to the reader. I think adding the stoichiometric predictions to this table will help put all of the predictions in one place, which is easier for the reader.

Page 6, line 116: I don't want to nit-pick too much, but 30 May to 1 June is not summer. That is the end of spring. 21 June is the official start of summer in the northern hemisphere.

Page 6, line 120: So, the fish were not measured before being placed in ethanol? All measurements were taken after preservation?

Page 7, line 139: More detail is needed here. How were the guts removed? Where did the authors cut? What landmarks did they use? Once it was out, did they uncoil it before photographing and measuring it? Was the liver removed with the gut (and the gall bladder and spleen, since they are usually attached to the gut)? That isn't clear at all. If the liver and other organs were removed from the body, then they cannot influence body stoichiometry. It isn't clear what body measurements were used? Standard length? Total length?

Page 7, line 147: How were gut contents removed? These are stomachless fish. So, did the authors stick with the proximal intestine in terms of content analysis? Did they include the whole gut? Distal intestine contents are much more difficult to identify.

Page 8, line 187: I am curious how the authors identified detritus? That is quite complicated and I don't feel like that was adequately shared. How did they differentiate digested animal material

for amorphous detritus? And, while on the topic of detritus, it is quite proteinaceous in comparison to algal material (Bowen et al. 1995; Wilson et al. 2003; Clements et al. 2017). So, although detritus can contain inorganic material, and a fair amount of water, both of which would dilute the quality, the microbial biomass (and biofilm) makes detritus quite nutritious. So, lumping algae and detritus together nutritionally is not completely fair: one (algae) is high in C and low in N, and the other the opposite (just dilute).

Bowen, S. H., Lutz, E. V., & Ahlgren, M. O. (1995). Dietary protein and energy as determinants of food quality: trophic strategies compared. *Ecology*, *76*, 899-907.

Clements, K. D., German, D. P., Piché, J., Tribollet, A., & Choat, J. H. (2017). Integrating ecological roles and trophic diversification on coral reefs: multiple lines of evidence identify parrotfishes as microphages. *Biological Journal of the Linnean Society*, *120*, 729-751. doi:10.1111/bij.12914

Wilson, S. K., Bellwood, D. R., Choat, J. H., & Furnas, M. J. (2003). Detritus in the epilithic algal matrix and its use by coral reef fishes. *Oceanography and Marine Biology Annual Review*, *41*, 279-309.

Page 14, line 305: no, this pattern is indeed not universal, especially for higher latitude herbivores. For instance, *Xiphister mucosus* has a range from California to the Bering Strait, and is herbivorous across the whole range (Horn et al. 1986; Hickerson and Cunningham 2005). It's sister taxon, *X. atropurpureus*, is equally omnivorous across the range. In lizards, the evolution of herbivory is negatively correlated with temperature, not positively (Espinoza et al. 2004).

Espinoza, R. E., Wiens, J. J., & Tracy, C. R. (2004). Recurrent evolution of herbivory in small, cold-climate lizards: Breaking the ecophysiological rules of reptilian herbivory. *Proceedings of the National Academy of Sciences of the United States of America*, *101*(48), 16819-16824. doi:10.1073/pnas.0401226101

Hickerson, M. J., & Cunningham, C. W. (2005). Contrasting Quaternary histories in an ecologically divergent sister pair of low-dispersing intertidal fish (*Xiphister*) revealed by multi-locus DNA analysis. *Evolution*, *59*, 344-360.

Page 14, line 312: Yes, and this speaks to the fish possibly feeding to meet their protein requirements, and thus, they have lower intake with more proteinaceous foods. Plus, more protein means longer time in the gut (Fris and Horn 1993; Horn et al. 1995). Protein digestion is pretty complex, especially without a stomach (and poeciliids lack one). Imagine yourself eating a large proteinaceous meal. You are full for some length of time afterwards and won't need another meal for some time. It is the same for the fish.

Fris, M. B., & Horn, M. H. (1993). Effects of diets of different protein content on food consumption, gut retention, protein conversion, and growth of *Cebidichthys violaceus* (Girard), an herbivorous fish of temperate zone marine waters. *Journal of Experimental Marine Biology and Ecology*, *166*, 185-202.

Horn, M. H., Mailhiot, K. F., Fris, M. B., & McClanahan, L. L. (1995). Growth, consumption, assimilation and excretion in the marine herbivorous fish *Cebidichthys violaceus* (Girard) fed natural and high protein diets. *Journal of Experimental Marine Biology and Ecology*, 190, 97-108.

Page 16, line 348: Are there no other stoichiometric studies of *Gambusia*? Perhaps in the native range? Any known differences between *Gambusia affinis* and *Gambusia holbrooki*? Looks like the authors already cite the Pyke papers.

Page 16, line 364: Here, the authors state, “Though decreasing condition factor was significantly related to increasing temperature in NZ where plant-based diets were common and in CA where animal-based diets were common, suggesting a strong effect of temperature on fish condition regardless of diet.” I feel like temperature affects on CF are well known, especially for aquaculture species and that there is usually a negative relationship with increasing temperature, especially if the diet is lacking in any way, or if the temperature gets passed their optimum. Thus, I don’t think this result is novel for *Gambusia* and the authors should find that relevant literature. For example:

<https://www.sciencedirect.com/science/article/pii/S2352513418300905>

What is the normal temperature range for *G. affinis*? What is their optimal temperature?

Table 4: I don’t like that M is used for male and for body mass. Maybe call body mass M_b . It just clarifies it for the reader.

Appendix B

Below is the feedback from the associated editor and reviewers and our responses coded in blue.

Associate Editor

Board Member: 1

Comments to Author:

This study investigates changes in diet, morphology, and elemental composition among populations of mosquitofish from two replicated temperature gradients. Both reviewers and myself found that the study's questions are interesting and the study to be well-conducted. Changes in diet in response to changes in climate, including temperature shifts, are critical to understand in our changing world, and data from natural systems are important.

Both reviewers raise important points that should be addressed.

First, the authors describe the observed variation in mosquitofish traits as a response to temperature. As reviewer 1 points out, other environmental variables are likely to vary among sites and to covary with temperature, so it is critical to evaluate some of these to allow a stronger assessment of the link to temperature. How do the conductivity and DO data that the authors collected covary with temperature and relate to the measured traits? Does the presence and abundance of predators and competitors vary among sites, and does this contribute to explaining trait variation? Does temperature have more explanatory power than a model of purely spatial variation among sites (temperature and space are likely to covary)? I would like to see the authors bring in additional data and modelling to address these possibilities.

Second, and related to the above point, more information is needed about the sites, as reviewer 1 notes. Is there any evidence that they are genetically differentiated populations? Without knowing this, it's hard to evaluate the correlations with temperature among sites. If closely-related and – located sites have been included, they will tend to cluster and might inflate the strength of relationships between fish traits and environmental variables. Is there additional temperature data beyond the authors' point measures to support the inference of long-term consistent temperature variation among sites?

Finally, the authors contrast the hypotheses that shifts in temperature will affect diet through affecting either organism metabolism or resource abundance. They conclude that changes in diet are more consistent with resource abundance shifts. But both hypotheses might be operating concurrently, so we need a discussion of whether mosquitofish metabolism might also shifted in response to temperature before concluding that it's mainly changing resource abundance that's important.

Reviewer 2 has suggested additional literature to set context and both reviewers have made additional suggestions that I hope the authors will find helpful.

We have made all changes suggested by yourself and both reviewers. Below, the key changes are discussed in regards to your comments above.

First, we now include data on each site's physiochemical composition (pH, dissolved oxygen, specific conductivity), site physical distance, and the presence of competitors or predators (Table 2, Table S1). Our site locations are also shown on a map (Fig S1), and the GPS co-ordinates of the sites are listed in the supplementary materials (Table S1). To understand if physiochemical variables were correlated with temperature, we now include correlation analyses to determine the relationship among physiochemical variables (Table S5). Further, to disentangle the role of spatial proximity and temperature, we performed a Mantel test that confirmed no correlation between site physical distance and site temperature difference (Table S2). We now present the known competitor or

predator species at each site in Table 2; other fish presence is not confounded with temperature across our study systems.

Second, though we do not have evidence that the populations are genetically different, we have added more information about our study systems into the main text, including that they are not hydrologically connected. Further, as above, the spatial separation of sites was not related to temperature, and as such, spatial clustering issues should be minimal. We have now included information on average site temperature and temperature variation where this is available (Table 2). Temperature does vary within sites, but the range is not large compared to the among site temperature range.

Third, we clarify in the conclusion our expectations around metabolic rate influencing dietary patterns. Finally, we have included references and clarified statements according to reviewer two's comments.

Referee: 1

Comments to the Author(s)

This paper has the focus on two important aspects of our constantly changing world: invasive species and rising temperatures. As such, I think it will be of high general interest for a wide readership. Nonetheless, I have identified some aspects that the authors need to address as part of a revision.

Main comments:

1 - General: The authors begin by making a strong link towards increasing temperatures and how this will affect organisms. While I do not disagree with this link, I think it needs to be a bit more nuanced, and more carefully developed as the current data does not look at changing temperatures, but at different (potentially static) temperatures at discrete sites. These sites are likely to also differ in a whole slew of other biotic and abiotic variables besides temperature (please see additional points below), so the direct link from those sites to changing temperature on a wider scale is not very apparent to me in the current manuscript.

We have edited the introduction and removed referral to changing temperatures as suggested. We also added a "space for time substitution" statement into the introduction and into the discussion to clarify that we are not looking at changing temperatures (lines 86 and 378).

2 - Methods, lines 102-115: Much more information on the actual study systems is needed. How far are they apart, i.e., do they constitute true independent samples; also, a map - even if just for the supplementary material - would be extremely helpful. The authors cannot expect all readers to try and look up all the previous studies on these populations you cited. At the very least, I suggest adding latitude and longitude to Table 1. I also suggest to add a figure to the supplement that shows maps for the relevant regions in California and NZ, with markers identifying the location of each site. I tried getting the information I needed from some of the cited papers, but could not find all of the relevant information even in previous papers that are cited.

The study sites within each country are quite far apart (1-154 km to nearest site) and they are not hydrologically connected. We have clarified this in the manuscript (Line 114). We have also added a figure (Figure S1) which shows the site locations and a table with GPS co-ordinates in addition to distances to the closest- and furthest sites (Table S1). We have added reference to the site figure and table in the main text on line 115. To ensure that temperature was not confounded with distance, we performed a Mantel test to examine correlation between spatial proximity and temperature difference across sites within NZ and CA separately. We found no significant relationship (lines 115-116 and Table S2).

3 - Methods, lines 117-118: Based on this information here and in Table 1, these seem to be point measurements. How reliable are they relative to diel variation, or even weekly and monthly variation. Do the authors have longer-term data on these sites as well, that provide some evidence for temporal stability? Phenotypes of the fish the authors collected are likely to represent environmental conditions experienced over a longer period prior to collection. How can the authors be sure that their single-point measurements are properly representing these longer-term patterns? We have temperature data from multiple time points for most, but not all of our sites. We have added average temperature and temperature variation data to Table 2 where this was available for our sites. Temperature does vary within sites, but the range is not large compared to the temperature range across sites. In addition, temperature on the date of *Gambusia* collection and average data temperatures over time were highly correlated in NZ ($r = 0.91$) and in CA ($r = 0.95$), suggesting that the temperatures at the time of collection are likely to be representative of longer-

term environmental conditions. We have retained the use of temperature on the date of fish collection (as opposed to long-term averages) in our analyses as gut contents in the fish will reflect feeding over the past few days at most in these fish.

4 - Methods, same lines as before: The authors mentioned at the beginning of the methods that these sites are all in geothermal springs. Do they have data on other water parameters in these aquatic habitats that could be due to different composition of discharge in each spring? I.e., amount of dissolved solids, minerals/salts, turbidity, any stressful compounds, etc.? Without this knowledge it will be difficult to really assess whether the uncovered patterns are really due to temperature or could be related to other, non-assessed differences between habitats. Following on from this, the authors further state that besides temperature, also conductivity and DO were quantified; yet, they seem not to have been included in any analytical models? I would like to see some rationale for this somewhere (I apologise if I missed it).

We do not have detailed data regarding chemical composition of the water. Specific conductivity gives us a rough idea of chemical diversity among sites as it reflects total ionic solutes. With the exception of one site, PK, specific conductance is not particularly high compared to non-geothermal systems (e.g., 0.095 – 0.875 mS/cm; Moffett et al 2015 *Freshwater Biology* 60, 1671-1687), suggesting these sites are not chemically extreme. We have also added pH data which we have for most of our sites. The values are generally circumneutral, suggesting that chemical conditions are not particularly extreme or variable among most of our sites. Dissolved oxygen is also generally high across our sites, particularly considering *Gambusia* are not overly sensitive to dissolved oxygen (they behaviorally respire at the freshwater-air interface when oxygen concentrations are low). Temperature is expected to correlated with dissolved oxygen and conductivity to some degree, although we used % saturation and specific conductance to minimize that issue. A full correlation table has been added to the supplementary materials (Table S5), which shows no significant relationships between temperature and the other physiochemical variables. We have provided water chemistry variables in text (Table 2) to set context and haven't added them to the model because there is no apparent mechanistic link between them and fish diet or morphology. We have adjusted our text to highlight that temperature was not correlated with other physiochemical variables measured (lines 111-114).

5 - Methods, lines 183-184: However, even relative gut length often differs between the sexes in *Gambusia*, other poeciliids and other fishes (e.g. Scharnweber et al. 2011 *Bull Fish Biol* 13:11-20, Näslund 2018 *Fishes* 3:38); in other words, even accounting for differences in body size between males and females, females may still have larger relative gut lengths. I therefore think sex will also have to be included as a factor in this analyses; or analyses should be split by sex.

Thank you for highlighting this - we have included sex as a factor in the analyses. As you hypothesized females in both NZ and CA had greater relative gut lengths. Notably, females and males show the same pattern consistent with that we observed with both sexes combined. We have updated Fig 1 c and f to reflect the variation in relative gut length between sexes.

6 - Discussion, lines 303-305: What about the role of competitors and predators? This also links back to my comment #1 on more detail on characteristics and location of each site. Are there any species of similar niche in these habitats, e.g. any pupfish in CA? Any native species in NZ that could influence *Gambusia* responses? Piscivorous fishes? If so, how could the presence or absence of competitors or predators have influenced dietary and morphological patterns (e.g., eye size discussion in lines 337-340). I think this needs to be more properly addressed.

We have added to Table 2 the species present in the sites which could compete with or predate on *Gambusia*. The presence of other fishes is not confounded with temperature, which we mention on lines 127 to 128. We have recently done comprehensive analysis of morphology in a common garden rearing experiment that is showing countergradient evolution of body shape in response to

temperature in the California populations so we are fairly confident that temperature is a key factor in morphology differences across sites.

7 - Discussion, lines 362-367: The condition factor only evaluates the weight/mass to length relationship, but not really fat storage. As such, you could have a situation in which fish have a different body shape (resulting in a different condition factor) but the same level of body fat. This needs to at least be acknowledged here.

This is a good point, and we now acknowledge it in our discussion that differences in fish condition factor may not be related to fat storage (lines 370 – 373).

Specific comments:

Introduction, line 61: Please delete the comma after 'If' at the beginning of the sentence.

Removed comma.

Methods, line 149: I am not a native speaker but this does not sound correct. Did the authors mean 'The proportion that each'?

Edited this sentence as suggested (Line 155).

Ethics Statement: What about a sampling permit for NZ? I know that Gambusia are invasive species there, but one usually still needs an official permit to specifically collect them (even if just in case the collecting effort might influence native species adversely)?

We had all the relevant collection and biosecurity permits for working with the fish in both countries. We have added this information to the ethics statement.

Tables 3 and 4: Since the actual p-values are provided here, the stars denoting the level of significance are not needed and redundant.

We have removed the stars from Table 3, where p-values are also included.

References, lines 485-486: Information for reference #11 seems incomplete.

References, lines 510-512, 515-517, 563-564, 573-576: Please write out the journal name.

We have edited the journal names as suggested.

Referee: 2

Review of "Consumer trait responses track different resource changes along replicated thermal gradients" by Moffett et al., a manuscript submitted to PRSB.

In this paper, the authors studied how *Gambusia affinis* responded to changing temperature and resource availability across thermal gradients in California and New Zealand. Thus, they represented replicate thermal gradient experiments with separate invasions of *G. affinis* in the two locations (*Gambusia* are not native to either location and had been in each for at least 85 years or so). They made some predictions about how temperature would affect diet (increased algal consumption with increased temperature) and how these changes would in turn affect body morphology (including mouth and eye shape, and gut length). Their hypotheses basically underscored that the fish would need more C due to higher metabolic rates at higher temperatures. Overall, I have no problems with the science itself. It is well-designed and executed. I do have problems with the scholarship, which is insufficient in many areas of the introduction. And, it is unfortunate that they could not replicate their findings in NZ in CA, showing that the pattern in NZ is not a universal one (mainly because the resource changes with temperature were not uniform among the locations). The variation in the responses by the fish are interesting in and of themselves, but I wonder if they are enough for PRSB. The introduction needs some work to match the citations with what is stated in the sentences. They don't always match up. The idea of complementarity isn't necessarily a new one, as omnivores are known to seek out C sources for energy, and more proteinaceous sources for N:

Raubenheimer, D., Zemke-White, W. L., Phillips, R. J., & Clements, K. D. (2005). Algal macronutrients and food selectivity by the omnivorous marine fish *Girella tricuspidata*. *Ecology*, *86*, 2601-2610.

And, the idea that animals would eat more of a low-protein diet to meet their needs isn't entirely novel, either. So, I appreciate many aspects of this study and it was an amazing effort in multiple locations. I applaud all of that. But, the writing left me unsatisfied.

Below, I detail comments on many parts of the introduction where the citations don't match the information in the sentences.

Page 3, line 47: Here, the authors can cite McWilliams and Karasov (2014) Proc Royal Soc B, where they used cold temperature to increase metabolic demand in an endotherm (bird). The birds showed increased food intake and changes in gut structure and function (which is key since most of the other cited studies didn't look at changes to the digestive system).

McWilliams, S. R., & Karasov, W. H. (2014). Spare capacity and phenotypic flexibility in the digestive system of a migratory bird: defining the limits of animal design. *Proceedings of the Royal Society B: Biological Sciences*, *281*(1783), 20140308. doi:doi:10.1098/rspb.2014.0308

We have added this citation as suggested (line 45).

Page 3, line 50: The authors cite Floeter et al. as evidence of some temperature-based constraints on digestive physiology of herbivores. Note that Floeter et al. provide no evidence of any such constraints, and in fact, Dr. Kendall Clements (University of Auckland) has a research program that includes many cold-water herbivores (e.g., *Odax pullus*) that have rates of gastrointestinal fermentation that rival endotherms at cooler temperatures (Mountfort et al. 2002). Not to mention, no impedance of growth (Trip et al. 2014) or diet shifts (Johnson et al. 2020) at higher latitudes.

Antarctic omnivores graze on algae from the bottom of the ice (Iken et al. 1997), and there are cool temperate herbivores in the northern hemisphere that digest algae with high efficiency as well (Horn et al. 1986; German et al. 2015). So, the authors should re-think that part of this sentence. The larger patterns of lower diversity and abundance of herbivorous fishes at higher latitudes can have many explanations beyond a digestive constraint (e.g., reproductive limitations, or stoichiometric ones).

German, D. P., Sung, A., Jhaveri, P. K., & Agnihotri, A. (2015). More than one way to be an herbivore: convergent evolution of herbivory using different digestive strategies in prickleback fishes (family

Stichaeidae). *Zoology*, 118, 161-170.

Horn, M. H., Neighbors, M. A., & Murray, S. N. (1986). Herbivore responses to a seasonally fluctuating food supply: growth potential of two temperate intertidal fishes based on the protein and energy assimilated from their macroalgal diets. *Journal of Experimental Marine Biology and Ecology*, 103, 217-234.

Iken, K., Barrera-Oro, E. R., Quartino, M. L., Casaux, R. J., & Brey, T. (1997). Grazing by the Antarctic fish *Notothenia coriiceps*: evidence for selective feeding on macroalgae. *Antarctic Science*, 9, 386-391.

Johnson, J. S., Raubenheimer, D., Bury, S. J., & Clements, K. D. (2020). Does temperature constrain diet choice in a marine herbivorous fish? *Marine Biology*, 167(7), 99. doi:10.1007/s00227-020-3677-z

Trip, E. D. L., Clements, K. D., Raubenheimer, D., & Choat, J. H. (2014). Temperature-related variation in growth rate, size, maturation and life span in a marine herbivorous fish over a latitudinal gradient. *Journal of Animal Ecology*, 83(4), 866-875. doi:10.1111/1365-2656.12183

We have removed part of this sentence as suggested.

Page 3, line 52: The authors state here that it could be C:N ratios that drive dietary changes with increasing metabolic rate, then cite two papers that looked at C:P (and number 9 specifically in animals with high P content), not C:N. Please find citations that correctly show this for C:N ratios.

We have edited the sentence to refer to phosphorous and not nitrogen (line 48).

Page 3, line 52: The authors state here that "Alternatively, consumers may shift away from a plant-rich diet towards an animal-rich diet due to the energetic inefficiency of digesting plant materials". They cite a paper on carnivorous trout (4), a model (number 10), a review article (11) that doesn't really cover inefficient digestion of plant material in herbivores (in fact, it shows herbivores use plants pretty well), and an article that used synthetic diets with varying protein to carbohydrate ratios in grasshoppers. None of these citations match this sentence to show energetic inefficiency of digesting plant materials by animals that normally consume plants (although Pandas don't do much with the fiber portion of bamboo; Dierenfeld et al. 1982; Nie et al. 2019). I ask the authors to dig and find such papers, because there are few (Floeter cites one on reptiles that is highly flawed).

Dierenfeld, E. S., Hintz, H. F., Robertson, J. B., Van Soest, P. J., & Oftedal, O. T. (1982). Utilization of bamboo by the giant panda. *Journal of Nutrition*, 112, 636-641.

Nie, Y. G., Wei, F. W., Zhou, W. L., Hu, Y. B., Senior, A. M., Wu, Q., . . . Raubenheimer, D. (2019). Giant Pandas Are Macronutritional Carnivores. *Current Biology*, 29(10), 1677-1682.e1672.

doi:<https://doi.org/10.1016/j.cub.2019.03.067>

We have updated the citations to match the information in this sentence (lines 48-50).

Page 4, line 67: Here, the authors state that "For example, plant-rich diets are associated with longer gut lengths within and across species due to the greater processing times required to process refractory materials in plants [11, 23, 24]." They cite a review and two papers that examine gut length in fishes, yet none of them measure retention time of food in the gut (e.g., see Horn 1989; German 2011 for discussions on this) or digestibility of refractory plant material. The authors imply that herbivores hold food in their guts longer and that increased retention time (i.e., decreased transit rate) is associated with longer guts to allow microbes to help in fiber digestion. This is not true across the board (Horn 1989; Clements et al. 2014), and there are few papers that actually measure fiber digestion in fishes (German 2009 is an example). A longer gut simply means higher intake (particularly in non-ruminant animals, which is most animals, and this is covered in Karasov and Douglas 2013 and Karasov and Martinez del Rio 2007. The authors already cite Leigh et al. 2018, which actually has a detailed discussion of this). Higher intake is associated with more rapid gut transit, not slower transit (German 2011), especially in thin walled intestines, like those of poeciliids. So, the authors here confuse "processing time" with intake. Unless the gut has some mechanism to

slow transit (valves, ceca, compartmentalization), then longer guts usually mean rapid transit (and actually, less microbial involvement in the digestion of plant refractory material). Few fishes really have long transit and a reliance on microbes to digest "refractory" material in plants and algae. In fact, those fishes with rapid fermentation occurring in their hindguts may be targeting mannitol and not the more fibrous parts of algae (White et al. 2010; Clements et al. 2014). So, the authors cannot make this statement and cite papers that do not show what they say. Longer guts mean higher intake. That's it. Herbivores can take many different strategies to digest plant material (German et al. 2015). Higher intake and longer gut usually means they are going after more soluble portions of the plant material and are passing more of it through their gut per unit time to meet their nutritional requirements (perhaps for protein; Horn et al. 1986; Clements and Raubenheimer 2006). Finally, no poeciliids have been shown to have much in the way of gastrointestinal fermentation occurring, so a longer gut in them likely just equates to higher intake and more rapid gut transit, as described above. The reported relative gut lengths are short for animals "reliant on plant material". I share all of this detail because these are issues with the writing, not the science performed. So, I want the authors to understand why gut length is a different issue than what they shared. High intake is probably about meeting their protein requirement from a diet that is relatively low in it. Finally, I share all of these citations not for all of them to be cited, but for the authors to have access to the information to give context to the issue at hand.

Clements, K. D., & Raubenheimer, D. (2006). Feeding and nutrition. In D. H. Evans (Ed.), *The physiology of fishes* (pp. 47-82). Boca Raton, FL: CRC Press.

Clements, K. D., Angert, E. R., Montgomery, W. L., & Choat, J. H. (2014). Intestinal microbiota in fishes: what's known and what's not. *Molecular Ecology*, 23(8), 1891-1898. doi:10.1111/mec.12699

German, D. P. (2009). Inside the guts of wood-eating catfishes: can they digest wood? *Journal of Comparative Physiology B*, 179, 1011-1023.

German, D. P. (2011). Digestive Efficiency. In A. P. Farrel (Ed.), *Encyclopedia of Fish Physiology: From Genome to Environment* (Vol. 3, pp. 1596-1607). San Diego: Academic Press.

Horn, M. H. (1989). Biology of Marine Herbivorous Fishes. *Oceanography and Marine Biology Annual Review*, 27, 167-272.

White, W. L., Coveny, A. H., Robertson, J., & Clements, K. D. (2010). Utilisation of mannitol by temperate marine herbivorous fishes. *Journal of Experimental Marine Biology and Ecology*, 391(1-2), 50-56. doi:http://dx.doi.org/10.1016/j.jembe.2010.06.007

We have corrected this sentence as you suggested as;

"For example, plant-rich diets are associated with longer gut lengths due to higher food intake rates". on line 62-63.

Page 4, lines 73-75: are the authors suggesting that in a single species, these morphological changes (downward turned vs upward turned mouths) occur across a thermal gradient in the same species? Or that these changes can occur rapidly with climate change? It isn't clear.

We have clarified this sentence as;

"Thus, if metabolic demand and resource availability change along thermal gradients, the resulting changes in resource use may be associated with adaptive changes to morphological traits." on line 68-70.

Page 5, line 95: The authors need to cite Table 2 here so that the predictions are clear to the reader. I think adding the stoichiometric predictions to this table will help put all of the predictions in one place, which is easier for the reader.

We now cite Table 1 and we have added stoichiometric predictions to this table as suggested.

Page 6, line 116: I don't want to nit-pick too much, but 30 May to 1 June is not summer. That is the end of spring. 21 June is the official start of summer in the northern hemisphere.

We have removed the term summer from this sentence (line 122).

Page 6, line 120: So, the fish were not measured before being placed in ethanol? All measurements were taken after preservation?

All fish measurements were done on thawed fish.

Page 7, line 139: More detail is needed here. How were the guts removed? Where did the authors cut? What landmarks did they use? Once it was out, did they uncoil it before photographing and measuring it? Was the liver removed with the gut (and the gall bladder and spleen, since they are usually attached to the gut)? That isn't clear at all. If the liver and other organs were removed from the body, then they cannot influence body stoichiometry. It isn't clear what body measurements were used? Standard length? Total length?

We have added more detail as requested to our methods for clarity on lines 143-147. Further, we clarify in Table 2 that standard length was used.

Page 7, line 147: How were gut contents removed? These are stomachless fish. So, did the authors stick with the proximal intestine in terms of content analysis? Did they include the whole gut? Distal intestine contents are much more difficult to identify.

We used the entire gut. We clarify this on lines 143-144.

Page 8, line 187: I am curious how the authors identified detritus? That is quite complicated and I don't feel like that was adequately shared. How did they differentiate digested animal material for amorphous detritus? And, while on the topic of detritus, it is quite proteinaceous in comparison to algal material (Bowen et al. 1995; Wilson et al. 2003; Clements et al. 2017). So, although detritus can contain inorganic material, and a fair amount of water, both of which would dilute the quality, the microbial biomass (and biofilm) makes detritus quite nutritious. So, lumping algae and detritus together nutritionally is not completely fair: one (algae) is high in C and low in N, and the other the opposite (just dilute).

Bowen, S. H., Lutz, E. V., & Ahlgren, M. O. (1995). Dietary protein and energy as determinants of food quality: trophic strategies compared. *Ecology*, 76, 899-907.

Clements, K. D., German, D. P., Piché, J., Tribollet, A., & Choat, J. H. (2017). Integrating ecological roles and trophic diversification on coral reefs: multiple lines of evidence identify parrotfishes as microphages. *Biological Journal of the Linnean Society*, 120, 729-751. doi:10.1111/bij.12914

Wilson, S. K., Bellwood, D. R., Choat, J. H., & Furnas, M. J. (2003). Detritus in the epilithic algal matrix and its use by coral reef fishes. *Oceanography and Marine Biology Annual Review*, 41, 279-309.

We now describe how we classified detritus in our methods on lines 160-161.

Page 14, line 305: no, this pattern is indeed not universal, especially for higher latitude herbivores. For instance, *Xiphister mucosus* has a range from California to the Bering Strait, and is herbivorous across the whole range (Horn et al. 1986; Hickerson and Cunningham 2005). It's sister taxon, *X. atropurpureus*, is equally omnivorous across the range. In lizards, the evolution of herbivory is negatively correlated with temperature, not positively (Espinosa et al. 2004).

Espinosa, R. E., Wiens, J. J., & Tracy, C. R. (2004). Recurrent evolution of herbivory in small, cold-climate lizards: Breaking the ecophysiological rules of reptilian herbivory. *Proceedings of the National Academy of Sciences of the United States of America*, 101(48), 16819-16824. doi:10.1073/pnas.0401226101

Hickerson, M. J., & Cunningham, C. W. (2005). Contrasting Quaternary histories in an ecologically divergent sister pair of low-dispersing intertidal fish (*Xiphister*) revealed by multi-locus DNA analysis. *Evolution*, 59, 344-360.

Interesting example, we now mention in our discussion (line 307) that this pattern is common outside of our dataset.

Page 14, line 312: Yes, and this speaks to the fish possibly feeding to meet their protein requirements, and thus, they have lower intake with more proteinaceous foods. Plus, more protein means longer time in the gut (Fris and Horn 1993; Horn et al. 1995). Protein digestion is pretty complex, especially without a stomach (and poeciliids lack one). Imagine yourself eating a large proteinaceous meal. You are full for some length of time afterwards and won't need another meal for some time. It is the same for the fish.

Fris, M. B., & Horn, M. H. (1993). Effects of diets of different protein content on food consumption, gut retention, protein conversion, and growth of *Cebidichthys violaceus* (Girard), an herbivorous fish of temperate zone marine waters. *Journal of Experimental Marine Biology and Ecology*, 166, 185-202.

Horn, M. H., Mailhiot, K. F., Fris, M. B., & McClanahan, L. L. (1995). Growth, consumption, assimilation and excretion in the marine herbivorous fish *Cebidichthys violaceus* (Girard) fed natural and high protein diets. *Journal of Experimental Marine Biology and Ecology*, 190, 97-108.

Reliance on high protein foods at warmer temperatures likely resulted in a shorter gut length – as you suggest. Though, it is still interesting that the relative importance of dietary items did not predict gut length in California, which we discuss on lines 314-318 (Fig S5).

Page 16, line 348: Are there no other stoichiometric studies of *Gambusia*? Perhaps in the native range? Any known differences between *Gambusia affinis* and *Gambusia holbrooki*? Looks like the authors already cite the Pyke papers.

We could not find any other stoichiometric studies from *Gambusia*, we mention this in the main text on lines 348-351.

Page 16, line 364: Here, the authors state, "Though decreasing condition factor was significantly related to increasing temperature in NZ where plant-based diets were common and in CA where animal-based diets were common, suggesting a strong effect of temperature on fish condition regardless of diet." I feel like temperature affects on CF are well known, especially for aquaculture species and that there is usually a negative relationship with increasing temperature, especially if the diet is lacking in any way, or if the temperature gets passed their optimum. Thus, I don't think this result is novel for *Gambusia* and the authors should find that relevant literature.

For example:

<https://www.sciencedirect.com/science/article/pii/S2352513418300905>

We now discuss and cite other papers, including the paper cited above, which show a reduction in condition factor at warmer temperatures is common (line 370).

What is the normal temperature range for *G. affinis*? What is their optimal temperature?

We have added this information into the manuscript, lines 103-104.

Table 4: I don't like that M is used for male and for body mass. Maybe call body mass Mb. It just clarifies it for the reader.

We have changed 'M' to Mass for clarity.

Appendix C

Below is the feedback from the associated editor and reviewers and our responses coded in blue.

Associate Editor Board Member: 1
Comments to Author:

Both original reviewers and myself found the manuscript improved, and the additional information about site location, average temperature and other properties is helpful. However, it remains my view that additional data and modelling are needed before the results can be seen as a fully convincing case that temperature drives population variation in the mosquitofish traits studied here.

First, a model comparison approach (e.g. based on Akaike information criterion) would help to evaluate whether a model including temperature has more explanatory power than a purely spatial model, or models based on pH, other chemical properties, or predators and competitors. A null model (e.g. that includes only the intercept) should also be included in the comparison set. The authors have introduced a Mantel test to show that temperature and spatial differences among sites are not strongly correlated, but this does not exclude the possibility that spatial variation explains trait variation better than temperature. Second, I suggest the authors consider an alternative spatial analysis to the Mantel test (see Legendre et al. 2015, Should the Mantel test be used in spatial analysis? *Methods in Ecology and Evolution* 6: 1239-1247).

Thank you for your suggestions. We have carried out a thorough model comparison approach for all traits in our manuscript. We now have six models for each trait, each with a different dependent factor (null, temperature, dissolved oxygen, specific conductivity, competitor presence, and spatial variation) (Tables 3, S5, S6). For a more robust method of quantifying spatial variation, we used distance-based Moran's eigenvector maps to reduce spatial variation into eigenfunctions. These models are referred to throughout our results. We found that temperature was an important driver of our key traits (gut length, %C, %N), but we acknowledge that other environmental drivers better explained other traits.

Third, I would like to see a more robust analysis of predator and competitor effects, as Reviewer 1 raised in their initial review. The revised manuscript introduces some data on predator and competitor presence or absence, but predators and competitors are grouped together without clear distinction and only one predator or competitor is listed for a subset of sites (surely some sites must have had more than one?). No source is given for these data and there are no data on predator or competitor abundance. The authors state that predator or competitor presence is not associated with temperature, but there probably isn't a good way to evaluate this given the current quality of the predator/competitor data.

We have added more information on the heterospecific fishes into the manuscript, though reviewer one was satisfied with the changes we made after the initial review.

We now clarify in the text that the presence of other species was noted during site visits (Lines 117-118). In the supplement, we added a table describing the role of heterospecific fish as either competitors or predators and references supporting each of these classifications (Table S4). In some cases, the distinction between predator and competitor in our study system is unknown. Indeed, assigning other species as predators or competitors, in general, is in and of itself an assumption because direct evidence in our systems is lacking, as is the case in most studies. It is, for this reason, we refrained from defining their distinct role in our study system in Table 2. Our study systems were often isolated or too warm for many other species, so it is not surprising that there were no or only one heterospecific species present with *Gambusia*. This is one of the reasons why we chose them.

For completeness, we included the presence of heterospecific species that could potentially compete or predate on *Gambusia* into our correlation analysis to confirm that the presence of these species was not related to temperature. Finally, following your first discussion point above, we included predator or competitor presence as one of our key models in explaining trait variation. Together these two analyses provide a more robust analysis of predator and competitor effects.

Reviewer(s)' Comments to Author:

Referee: 1

Comments to the Author(s).

I would like to thank the authors for their careful revision and for the way they addressed my comments from the previous round of review (e.g., by adding maps of the study region to the supplement). I am satisfied with these changes, but when working through the new version, I stumbled over a few additional points that will need to be addressed.

Specific points:

Abstract, line 20: Maybe this is because I am not a native speaker, but 'Temperature rise' sounds odd to me and I would rephrase this to 'Rising temperatures'. Also, I think it should be 'demands' rather than 'demand'.

Changed 'Temperature rise' to 'Rising temperature' as suggested.

Abstract, line 30: I found the wording 'longer gut length to body length ratios' a bit odd, and would simply refer to this as 'relative gut length' as is done throughout the main text.

Edited as suggested.

Introduction, lines 45-50: Are the last two sentences supposed to explain the pattern found for *L. stagnalis* or refer back to the overall pattern? Currently, this is not fully clear from the wording. If the former, then the explanations seem to contradict the pattern reported as an example; if the latter, then please rephrase to make this connection more clear.

We mean the latter. Clarified as suggested.

Methods, lines 111-120: Given that these patterns are based on the measurements taken as part of this study, this should all be in past tense.

Changed to past tense as suggested.

Methods, line 127: Please change 'using' to 'via'

Changed.

Methods, lines 141-142: Here and for the other sample sizes below: Were these a mix of adult and juvenile fish or were these all adults? This would be important to know as a mixing of juveniles and adults introduces additional noise to the dataset and could result in spurious significance when comparing samples that could solely be based on the proportion of juveniles in different samples. I am assuming from the SL values provided in the data that all were adults but this would be important to state specifically.

We used mature or adult fish only. We now clarify this on lines 144 and 166.

Methods, line 188: Please provide a relevant citation for the sexual size dimorphism.
Citation added.

Results, line 229: Please add wording that makes clear that you do not consider this significant - the associated p-value shows this but your wording seems to suggest you considered this on the same level as the significant opposite trend in NZ.
Added.

Results, line 259: Please add 'and' prior to '%P'.
Added.

Table 4: The addition of the full stop after the t-value is confusing, given that a full stop is already part of every t-value. I suggest to simply not use any symbol for the '<0.05' significance. The bold already indicates it was significant, so no additional symbol is needed to separate it from *, ** and ***.
We have changed the symbols for clarity.

Figure 1: Detritus is mis-spelled on the y-axes of panels b and e.
Edited.

Referee: 2

Comments to the Author(s).

Review of "Consumer trait responses track change in resource supply along replicated thermal gradients", by Moffett et al., which has been resubmitted for potential publication in PRSB.

Overall, I am pleased with how the authors responded to the initial reviews. Both reviewers gave the authors significant things to change in the writing, and the authors responded appropriately (which is refreshing). In this second reading, I found a few terminology issues with the writing. These are simple "find and replace" kind of corrections. But, overall, I do find the story an interesting one.

Page 7, line 144: Pharynx, not throat. Throat is more of a mammalian (and really, human) term.

Changed to pharynx as suggested.

Page 8, lines 160-161: amorphous detritus is so much more than decomposing plant material. It is often hard to tell apart from mucus in the gut because it is brownish and has no discernable shapes. It is also richer in protein (Bowen et al. 1995). So, I do think the authors should acknowledge that they quantified degraded plant material (whether it was "predigested" by microbes in the environment or actually partially digested by the fish), and not detritus, per se. Amorphous detritus has a lot of microbial biomass in it (Bowen et al. 1995; Wilson et al. 2003).

Bowen, S. H., Lutz, E. V., & Ahlgren, M. O. (1995). Dietary protein and energy as determinants of food quality: trophic strategies compared. *Ecology*, 76, 899-907.

Wilson, S. K., Bellwood, D. R., Choat, J. H., & Furnas, M. J. (2003). Detritus in the epilithic algal matrix and its use by coral reef fishes. *Oceanography and Marine Biology Annual Review*, 41, 279-309.

Clarified sentence as suggested.

Page 16, line 332: mouth position moving ventrally or dorsally is odd terminology for fishes. Mouth placement is usually called "terminal" for mouths located at the direct tip of the head. Sub-terminal (or inferior) is what the authors are calling "ventrally", and "superior" or "supra terminal" is the terminology for what the authors are calling "dorsally". These are more of the "textbook" terms used in the literature. I do believe the authors don't wish to invent new terminology here.

Changed terminology throughout.

Page 17, lines 354-355: The authors are stating that *Gambusia* have determinant growth in males, but indeterminate growth in females. Sexual dimorphism in size is common in fishes, but I haven't heard that animals that generally have indeterminate growth (i.e., most fishes) actually "stop growing" at sexual maturity. It is more along the lines that growth slows more for males than for females. Please provide a citation that states that males have a "terminal" size and determinant growth, whereas females don't.

Clarified sentence.

Appendix D

Below is the feedback from reviewer one and our responses coded in blue.

Lines 104-106: Another very recent diet study that could be cited here is Pirroni et al. 2021 Ecol Evol 11:4379–4398.

We have added this citation.

Supplementary Tables: Thank you for changing how significance is indicated in Table 4 (now Table 5). Please also change the designation of significance in all the relevant supplementary tables, as again, using a full stop after the p-value to indicate significance level is very confusing.

We have changed the significance notation in the supplementary materials.